# Tight Complexity Bounds for Optimizing Composite Objectives

**Blake Woodworth**
Toyota Technological Institute at Chicago
Chicago, IL, 60637
blake@ttic.edu

**Nathan Srebro**
Toyota Technological Institute at Chicago
Chicago, IL, 60637
nati@ttic.edu

## Abstract

We provide tight upper and lower bounds on the complexity of minimizing the average of $m$ convex functions using gradient and prox oracles of the component functions. We show a significant gap between the complexity of deterministic vs randomized optimization. For smooth functions, we show that accelerated gradient descent (AGD) and an accelerated variant of SVRG are optimal in the deterministic and randomized settings respectively, and that a gradient oracle is sufficient for the optimal rate. For non-smooth functions, having access to prox oracles reduces the complexity and we present optimal methods based on smoothing that improve over methods using just gradient accesses.

## 1   Introduction

We consider minimizing the average of $m \geq 2$ convex functions:

$$\min_{x \in \mathcal{X}} \left\{ F(x) := \frac{1}{m} \sum_{i=1}^{m} f_i(x) \right\} \tag{1}$$

where $\mathcal{X} \subseteq \mathbb{R}^d$ is a closed, convex set, and where the algorithm is given access to the following gradient (or subgradient in the case of non-smooth functions) and prox oracle for the components:

$$h_F(x, i, \beta) = \left[ f_i(x), \ \nabla f_i(x), \ \mathrm{prox}_{f_i}(x, \beta) \right] \tag{2}$$

where

$$\mathrm{prox}_{f_i}(x, \beta) = \arg\min_{u \in \mathcal{X}} \left\{ f_i(u) + \frac{\beta}{2} \|x - u\|^2 \right\} \tag{3}$$

A natural question is how to leverage the prox oracle, and how much benefit it provides over gradient access alone. The prox oracle is potentially much more powerful, as it provides global, rather then local, information about the function. For example, for a single function ($m = 1$), one prox oracle call (with $\beta = 0$) is sufficient for exact optimization. Several methods have recently been suggested for optimizing a sum or average of several functions using prox accesses to each component, both in the distributed setting where each components might be handled on a different machine (e.g. ADMM [7], DANE [18], DISCO [20]) or for functions that can be decomposed into several "easy" parts (e.g. PRISMA [13]). But as far as we are aware, no meaningful lower bound was previously known on the number of prox oracle accesses required even for the average of two functions ($m = 2$).

The optimization of composite objectives of the form (1) has also been extensively studied in the context of minimizing empirical risk over $m$ samples. Recently, stochastic methods such as SDCA [16], SAG [14], SVRG [8], and other variants, have been presented which leverage the finite nature of the problem to reduce the variance in stochastic gradient estimates and obtain guarantees that dominate both batch and stochastic gradient descent. As methods with improved complexity, such

| | | L-Lipschitz | | γ-Smooth | |
| --- | --- | --- | --- | --- | --- |
| | | Convex, $\|x\| \leq B$ | λ-Strongly Convex | Convex, $\|x\| \leq B$ | λ-Strongly Convex |
| **Deterministic** Upper | | $\dfrac{mLB}{\epsilon}$ <br>**(Section 3)** | $\dfrac{mL}{\sqrt{\lambda\epsilon}}$ <br>**(Section 3)** | $m\sqrt{\dfrac{\gamma B^2}{\epsilon}}$ <br>(AGD) | $m\sqrt{\dfrac{\gamma}{\lambda}}\log\dfrac{\epsilon_0}{\epsilon}$ <br>(AGD) |
| **Deterministic** Lower | | $\dfrac{mLB}{\epsilon}$ <br>**(Section 4)** | $\dfrac{mL}{\sqrt{\lambda\epsilon}}$ <br>**(Section 4)** | $m\sqrt{\dfrac{\gamma B^2}{\epsilon}}$ <br>**(Section 4)** | $m\sqrt{\dfrac{\gamma}{\lambda}}\log\dfrac{\epsilon_0}{\epsilon}$ <br>**(Section 4)** |
| **Randomized** Upper | | $\dfrac{L^2B^2}{\epsilon^2}\wedge\left(m\log\dfrac{1}{\epsilon}+\dfrac{\sqrt{m}LB}{\epsilon}\right)$ <br>(SGD, A-SVRG) | $\dfrac{L^2}{\lambda\epsilon}\wedge\left(m\log\dfrac{1}{\epsilon}+\dfrac{\sqrt{m}L}{\sqrt{\lambda\epsilon}}\right)$ <br>(SGD, A-SVRG) | $m\log\dfrac{\epsilon_0}{\epsilon}+\sqrt{\dfrac{m\gamma B^2}{\epsilon}}$ <br>(A-SVRG) | $\left(m+\sqrt{\dfrac{m\gamma}{\lambda}}\right)\log\dfrac{\epsilon_0}{\epsilon}$ <br>(A-SVRG) |
| **Randomized** Lower | | $\dfrac{L^2B^2}{\epsilon^2}\wedge\left(m+\dfrac{\sqrt{m}LB}{\epsilon}\right)$ <br>**(Section 5)** | $\dfrac{L^2}{\lambda\epsilon}\wedge\left(m+\dfrac{\sqrt{m}L}{\sqrt{\lambda\epsilon}}\right)$ <br>**(Section 5)** | $m+\sqrt{\dfrac{m\gamma B^2}{\epsilon}}$ <br>**(Section 5)** | $m+\sqrt{\dfrac{m\gamma}{\lambda}}\log\dfrac{\epsilon_0}{\epsilon}$ <br>**(Section 5)** |

Table 1: Upper and lower bounds on the number of grad-and-prox oracle accesses needed to find $\epsilon$-suboptimal solutions for each function class. These are exact up to constant factors except for the lower bounds for smooth and strongly convex functions, which hide extra $\log\lambda/\gamma$ and $\log\sqrt{m\lambda/\gamma}$ factors for deterministic and randomized algorithms. Here, $\epsilon_0$ is the suboptimality of the point 0.

as accelerated SDCA [17], accelerated SVRG, and KATYUSHA [3], have been presented, researchers have also tried to obtain lower bounds on the best possible complexity in this settings—but as we survey below, these have not been satisfactory so far.

In this paper, after briefly surveying methods for smooth, composite optimization, we present methods for optimizing non-smooth composite objectives, which show that prox oracle access can indeed be leveraged to improve over methods using merely subgradient access (see Section 3). We then turn to studying lower bounds. We consider algorithms that access the objective $F$ only through the oracle $h_F$ and provide lower bounds on the number of such oracle accesses (and thus the runtime) required to find $\epsilon$-suboptimal solutions. We consider optimizing both Lipschitz (non-smooth) functions and smooth functions, and guarantees that do and do not depend on strong convexity, distinguishing between deterministic optimization algorithms and randomized algorithms. Our upper and lower bounds are summarized in Table 1.

As shown in the table, we provide matching upper and lower bounds (up to a log factor) for all function and algorithm classes. In particular, our bounds establish the optimality (up to log factors) of accelerated SDCA, SVRG, and SAG for randomized finite-sum optimization, and also the optimality of our deterministic smoothing algorithms for non-smooth composite optimization.

**On the power of gradient vs prox oracles** For non-smooth functions, we show that having access to prox oracles for the components can reduce the polynomial dependence on $\epsilon$ from $1/\epsilon^2$ to $1/\epsilon$, or from $1/(\lambda\epsilon)$ to $1/\sqrt{\lambda\epsilon}$ for $\lambda$-strongly convex functions. However, all of the optimal complexities for smooth functions can be attained with only component gradient access using accelerated gradient descent (AGD) or accelerated SVRG. Thus the worst-case complexity cannot be improved (at least not significantly) by using the more powerful prox oracle.

**On the power of randomization** We establish a significant gap between deterministic and randomized algorithms for finite-sum problems. Namely, the dependence on the number of components must be linear in $m$ for any deterministic algorithm, but can be reduced to $\sqrt{m}$ (in the typically significant term) using randomization. We emphasize that the randomization here is only in the algorithm—not in the oracle. We always assume the oracle returns an exact answer (for the requested component) and is *not* a stochastic oracle. The distinction is that the *algorithm* is allowed to flip coins in deciding what operations and queries to perform but the oracle must return an exact answer to that query (of course, the algorithm could simulate a stochastic oracle).

**Prior Lower Bounds** Several authors recently presented lower bounds for optimizing (1) in the smooth and strongly convex setting using component gradients. Agarwal and Bottou [1] presented a lower bound of $\Omega\left(m+\sqrt{\frac{m\gamma}{\lambda}}\log\frac{1}{\epsilon}\right)$. However, their bound is valid only for *deterministic* algorithms (thus not including SDCA, SVRG, SAG, etc.)—we not only consider randomized algorithms, but also show a much higher lower bound for deterministic algorithms (i.e. the bound of Agarwal

and Bottou is loose). Improving upon this, Lan [9] shows a similar lower bound for a restricted class of randomized algorithms: the algorithm must select which component to query for a gradient by drawing an index from a fixed distribution, but the algorithm must otherwise be deterministic in how it uses the gradients, and its iterates must lie in the span of the gradients it has received. This restricted class includes SAG, but not SVRG nor perhaps other realistic attempts at improving over these. Furthermore, both bounds allow only gradient accesses, not prox computations. Thus SDCA, which requires prox accesses, and potential variants are not covered by such lower bounds. We prove as similar lower bound to Lan's, but our analysis is much more general and applies to *any* randomized algorithm, making *any* sequence of queries to a gradient *and* prox oracle, and without assuming that iterates lie in the span of previous responses. In addition to smooth functions, we also provide lower bounds for non-smooth problems which were not considered by these previous attempts. Another recent observation [15] was that with access only to random component subgradients *without* knowing the component's identity, an algorithm must make $\Omega(m^2)$ queries to optimize well. This shows how relatively subtle changes in the oracle can have a dramatic effect on the complexity of the problem. Since the oracle we consider is quite powerful, our lower bounds cover a very broad family of algorithms, including SAG, SVRG, and SDCA.

Our deterministic lower bounds are inspired by a lower bound on the number of rounds of communication required for optimization when each $f_i$ is held by a different machine and when iterates lie in the span of certain permitted calculations [5]. Our construction for $m = 2$ is similar to theirs (though in a different setting), but their analysis considers neither scaling with $m$ (which has a different role in their setting) nor randomization.

**Notation and Definitions**   We use $\|\cdot\|$ to denote the standard Euclidean norm on $\mathbb{R}^d$. We say that a function $f$ is $L$-Lipschitz continuous on $\mathcal{X}$ if $\forall x, y \in \mathcal{X}$ $|f(x) - f(x)| \leq L \|x - y\|$; $\gamma$-smooth on $\mathcal{X}$ if it is differentiable and its gradient is $\gamma$-Lipschitz on $\mathcal{X}$; and $\lambda$-strongly convex on $\mathcal{X}$ if $\forall x, y \in \mathcal{X}$ $f_i(y) \geq f_i(x) + \langle \nabla f_i(x), y - x \rangle + \frac{\lambda}{2} \|x - y\|^2$. We consider optimizing (1) under four combinations of assumptions: each component $f_i$ is either $L$-Lipschitz or $\gamma$-smooth, and either $F(x)$ is $\lambda$-strongly convex or its domain is bounded, $\mathcal{X} \subseteq \{x : \|x\| \leq B\}$.

## 2   Optimizing Smooth Sums

We briefly review the best known methods for optimizing (1) when the components are $\gamma$-smooth, yielding the upper bounds on the right half of Table 1. These upper bounds can be obtained using only component gradient access, without need for the prox oracle.

We can obtain exact gradients of $F(x)$ by computing all $m$ component gradients $\nabla f_i(x)$. Running accelerated gradient descent (AGD) [12] on $F(x)$ using these exact gradients achieves the upper complexity bounds for deterministic algorithms and smooth problems (see Table 1).

SAG [14], SVRG [8] and related methods use randomization to sample components, but also leverage the finite nature of the objective to control the variance of the gradient estimator used. Accelerating these methods using the Catalyst framework [10] ensures that for $\lambda$-strongly convex objectives we have $\mathbb{E}\left[F(x^{(k)}) - F(x^*)\right] < \epsilon$ after $k = \mathcal{O}\left((m + \sqrt{\frac{m\gamma}{\lambda}}) \log^2 \frac{\epsilon_0}{\epsilon}\right)$ iterations, where $F(0) - F(x^*) = \epsilon_0$. KATYUSHA [3] is a more direct approach to accelerating SVRG which avoids extraneous log-factors, yielding the complexity $k = \mathcal{O}\left((m + \sqrt{\frac{m\gamma}{\lambda}}) \log \frac{\epsilon_0}{\epsilon}\right)$ indicated in Table 1.

When $F$ is not strongly convex, adding a regularizer to the objective and instead optimizing $F_\lambda(x) = F(x) + \frac{\lambda}{2} \|x\|^2$ with $\lambda = \epsilon/B^2$ results in an oracle complexity of $\mathcal{O}\left(\left(m + \sqrt{\frac{m\gamma B^2}{\epsilon}}\right) \log \frac{\epsilon_0}{\epsilon}\right)$.

The log-factor in the second term can be removed using the more delicate reduction of Allen-Zhu and Hazan [4], which involves optimizing $F_\lambda(x)$ for progressively smaller values of $\lambda$, yielding the upper bound in the table.

KATYUSHA and Catalyst-accelerated SAG or SVRG use only gradients of the components. Accelerated SDCA [17] achieves a similar complexity using gradient and prox oracle access.

## 3   Leveraging Prox Oracles for Lipschitz Sums

In this section, we present algorithms for leveraging the prox oracle to minimize (1) when each component is $L$-Lipschitz. This will be done by using the prox oracle to "smooth" each component,

and optimizing the new, smooth sum which approximates the original problem. This idea was used in order to apply KATYUSHA [3] and accelerated SDCA [17] to non-smooth objectives. We are not aware of a previous explicit presentation of the AGD-based deterministic algorithm, which achieves the deterministic upper complexity indicated in Table 1.

The key is using a prox oracle to obtain gradients of the $\beta$-Moreau envelope of a non-smooth function, $f$, defined as:

$$f^{(\beta)}(x) = \inf_{u \in \mathcal{X}} f(u) + \frac{\beta}{2} \|x - u\|^2 \tag{4}$$

**Lemma 1** ([13, Lemma 2.2], [6, Proposition 12.29], following [11]). *Let $f$ be convex and $L$-Lipschitz continuous. For any $\beta > 0$,*

1. *$f^{(\beta)}$ is $\beta$-smooth*

2. *$\nabla(f^{(\beta)})(x) = \beta(x - prox_f(x, \beta))$*

3. *$f^{(\beta)}(x) \leq f(x) \leq f^{(\beta)}(x) + \frac{L^2}{2\beta}$*

Consequently, we can consider the smoothed problem

$$\min_{x \in \mathcal{X}} \left\{ \tilde{F}^{(\beta)}(x) := \frac{1}{m} \sum_{i=1}^{m} f_i^{(\beta)}(x) \right\}. \tag{5}$$

While $\tilde{F}^{(\beta)}$ is *not*, in general, the $\beta$-Moreau envelope of $F$, it is $\beta$-smooth, we can calculate the gradient of its components using the oracle $h_F$, and $\tilde{F}^{(\beta)}(x) \leq F(x) \leq \tilde{F}^{(\beta)}(x) + \frac{L^2}{2\beta}$. Thus, to obtain an $\epsilon$-suboptimal solution to (1) using $h_F$, we set $\beta = L^2/\epsilon$ and apply any algorithm which can optimize (5) using gradients of the $L^2/\epsilon$-smooth components, to within $\epsilon/2$ accuracy. With the rates presented in Section 2, using AGD on (5) yields a complexity of $\mathcal{O}\left(\frac{mLB}{\epsilon}\right)$ in the deterministic setting. When the functions are $\lambda$-strongly convex, smoothing with a fixed $\beta$ results in a spurious log-factor. To avoid this, we again apply the reduction of Allen-Zhu and Hazan [4], this time optimizing $\tilde{F}^{(\beta)}$ for increasingly large values of $\beta$. This leads to the upper bound of $\mathcal{O}\left(\frac{mL}{\sqrt{\lambda\epsilon}}\right)$ when used with AGD (see Appendix A for details).

Similarly, we can apply an accelerated randomized algorithm (such as KATYUSHA) to the smooth problem $\tilde{F}^{(\beta)}$ to obtain complexities of $\mathcal{O}\left(m \log \frac{\epsilon_0}{\epsilon} + \frac{\sqrt{m}LB}{\epsilon}\right)$ and $\mathcal{O}\left(m \log \frac{\epsilon_0}{\epsilon} + \frac{\sqrt{m}L}{\sqrt{\lambda\epsilon}}\right)$—this matches the presentation of Allen-Zhu [3] and is similar to that of Shalev-Shwartz and Zhang [17].

Finally, if $m > L^2B^2/\epsilon^2$ or $m > L^2/(\lambda\epsilon)$, stochastic gradient descent is a better randomized alternative, yielding complexities of $\mathcal{O}(L^2B^2/\epsilon^2)$ or $\mathcal{O}(L^2/(\lambda\epsilon))$.

## 4 Lower Bounds for Deterministic Algorithms

We now turn to establishing lower bounds on the oracle complexity of optimizing (1). We first consider only deterministic optimization algorithms. What we would like to show is that for any deterministic optimization algorithm we can construct a "hard" function for which the algorithm cannot find an $\epsilon$-suboptimal solution until it has made many oracle accesses. Since the algorithm is deterministic, we can construct such a function by simulating the (deterministic) behavior of the algorithm. This can be viewed as a game, where an adversary controls the oracle being used by the algorithm. At each iteration the algorithm queries the oracle with some triplet $(x, i, \beta)$ and the adversary responds with an answer. This answer must be consistent with all previous answers, but the adversary ensures it is also consistent with a composite function $F$ that the algorithm is far from optimizing. The "hard" function is then gradually defined in terms of the behavior of the optimization algorithm.

To help us formulate our constructions, we define a "round" of queries as a series of queries in which $\lceil \frac{m}{2} \rceil$ distinct functions $f_i$ are queried. The first round begins with the first query and continues until exactly $\lceil \frac{m}{2} \rceil$ unique functions have been queried. The second round begins with the next query, and continues until exactly $\lceil \frac{m}{2} \rceil$ more distinct components have been queried in the second round, and so on until the algorithm terminates. This definition is useful for analysis but requires no assumptions about the algorithm's querying strategy.

## 4.1 Non-Smooth Components

We begin by presenting a lower bound for deterministic optimization of (1) when each component $f_i$ is convex and $L$-Lipschitz continuous, but is not necessarily strongly convex, on the domain $\mathcal{X} = \{x : \|x\| \le B\}$. Without loss of generality, we can consider $L = B = 1$. We will construct functions of the following form:

$$f_i(x) = \frac{1}{\sqrt{2}} |b - \langle x, v_0 \rangle| + \frac{1}{2\sqrt{k}} \sum_{r=1}^{k} \delta_{i,r} |\langle x, v_{r-1} \rangle - \langle x, v_r \rangle|. \tag{6}$$

where $k = \lfloor \frac{1}{12\epsilon} \rfloor$, $b = \frac{1}{\sqrt{k+1}}$, and $\{v_r\}$ is an orthonormal set of vectors in $\mathbb{R}^d$ chosen according to the behavior of the algorithm such that $v_r$ is orthogonal to all points at which the algorithm queries $h_F$ before round $r$, and where $\delta_{i,r}$ are indicators chosen so that $\delta_{i,r} = 1$ if the algorithm does *not* query component $i$ in round $r$ (and zero otherwise). To see how this is possible, consider the following truncations of (6):

$$f_i^t(x) = \frac{1}{\sqrt{2}} |b - \langle x, v_0 \rangle| + \frac{1}{2\sqrt{k}} \sum_{r=1}^{t-1} \delta_{i,r} |\langle x, v_{r-1} \rangle - \langle x, v_r \rangle| \tag{7}$$

During each round $t$, the adversary answers queries according to $f_i^t$, which depends only on $v_r, \delta_{i,r}$ for $r < t$, i.e. from previous rounds. When the round is completed, $\delta_{i,t}$ is determined and $v_t$ is chosen to be orthogonal to the vectors $\{v_0, ..., v_{t-1}\}$ as well as every point queried by the algorithm so far, thus defining $f_i^{t+1}$ for the next round. In Appendix B.1 we prove that these responses based on $f_i^t$ are consistent with $f_i$.

The algorithm can only learn $v_r$ after it completes round $r$—until then every iterate is orthogonal to it by construction. The average of these functions reaches its minimum of $F(x^*) = 0$ at $x^* = b \sum_{r=0}^{k} v_r$, so we can view optimizing these functions as the task of discovering the vectors $v_r$—even if only $v_k$ is missing, a suboptimality better than $b/(6\sqrt{k}) > \epsilon$ cannot be achieved. Therefore, the deterministic algorithm must complete at least $k$ rounds of optimization, each comprising at least $\lceil \frac{m}{2} \rceil$ queries to $h_F$ in order to optimize $F$. The key to this construction is that even though each term $|\langle x, v_{r-1} \rangle - \langle x, v_r \rangle|$ appears in $m/2$ components, and hence has a strong effect on the average $F(x)$, we force a deterministic algorithm to make $\Omega(m)$ queries during each round before it finds the next relevant term. We obtain (for complete proof see Appendix B.1):

**Theorem 1.** *For any $L, B > 0$, any $0 < \epsilon < \frac{LB}{12}$, any $m \ge 2$, and any deterministic algorithm $A$ with access to $h_F$, there exists a dimension $d = \mathcal{O}\left(\frac{mLB}{\epsilon}\right)$, and $m$ functions $f_i$ defined over $\mathcal{X} = \{x \in \mathbb{R}^d : \|x\| \le B\}$, which are convex and $L$-Lipschitz continuous, such that in order to find a point $\hat{x}$ for which $F(\hat{x}) - F(x^*) < \epsilon$, $A$ must make $\Omega\left(\frac{mLB}{\epsilon}\right)$ queries to $h_F$.*

Furthermore, we can always reduce optimizing a function over $\|x\| \le B$ to optimizing a strongly convex function by adding the regularizer $\epsilon \|x\|^2 / (2B^2)$ to each component, implying (see complete proof in Appendix B.2):

**Theorem 2.** *For any $L, \lambda > 0$, any $0 < \epsilon < \frac{L^2}{288\lambda}$, any $m \ge 2$, and any deterministic algorithm $A$ with access to $h_F$, there exists a dimension $d = \mathcal{O}\left(\frac{mL}{\sqrt{\lambda \epsilon}}\right)$, and $m$ functions $f_i$ defined over $\mathcal{X} \subseteq \mathbb{R}^d$, which are $L$-Lipschitz continuous and $\lambda$-strongly convex, such that in order to find a point $\hat{x}$ for which $F(\hat{x}) - F(x^*) < \epsilon$, $A$ must make $\Omega\left(\frac{mL}{\sqrt{\lambda \epsilon}}\right)$ queries to $h_F$.*

## 4.2 Smooth Components

When the components $f_i$ are required to be smooth, the lower bound construction is similar to (6), except it is based on squared differences instead of absolute differences. We consider the functions:

$$f_i(x) = \frac{1}{8} \left( \delta_{i,1} \left( \langle x, v_0 \rangle^2 - 2a \langle x, v_0 \rangle \right) + \delta_{i,k} \langle x, v_k \rangle^2 + \sum_{r=1}^{k} \delta_{i,r} \left( \langle x, v_{r-1} \rangle - \langle x, v_r \rangle \right)^2 \right) \tag{8}$$

where $\delta_{i,r}$ and $v_r$ are as before. Again, we can answer queries at round $t$ based only on $\delta_{i,r}, v_r$ for $r < t$. This construction yields the following lower bounds (full details in Appendix B.3):

**Theorem 3.** *For any $\gamma, B, \epsilon > 0$, any $m \geq 2$, and any deterministic algorithm $A$ with access to $h_F$, there exists a sufficiently large dimension $d = \mathcal{O}\big(m\sqrt{\gamma B^2/\epsilon}\big)$, and $m$ functions $f_i$ defined over $\mathcal{X} = \big\{x \in \mathbb{R}^d : \|x\| \leq B\big\}$, which are convex and $\gamma$-smooth, such that in order to find a point $\hat{x} \in \mathbb{R}^d$ for which $F(\hat{x}) - F(x^*) < \epsilon$, $A$ must make $\Omega\big(m\sqrt{\gamma B^2/\epsilon}\big)$ queries to $h_F$.*

In the strongly convex case, we use a very similar construction, adding the term $\lambda \|x\|^2 /2$, which gives the following bound (see Appendix B.4):

**Theorem 4.** *For any $\gamma, \lambda > 0$ such that $\frac{\gamma}{\lambda} > 73$, any $\epsilon > 0$, any $\epsilon_0 > \frac{3\gamma\epsilon}{\lambda}$, any $m \geq 2$, and any deterministic algorithm $A$ with access to $h_F$, there exists a sufficiently large dimension $d = \mathcal{O}\left(m\sqrt{\frac{\gamma}{\lambda}}\log\left(\frac{\lambda\epsilon_0}{\gamma\epsilon}\right)\right)$, and $m$ functions $f_i$ defined over $\mathcal{X} \subseteq \mathbb{R}^d$, which are $\gamma$-smooth and $\lambda$-strongly convex and where $F(0) - F(x^*) = \epsilon_0$, such that in order to find a point $\hat{x}$ for which $F(\hat{x}) - F(x^*) < \epsilon$, $A$ must make $\Omega\left(m\sqrt{\frac{\gamma}{\lambda}}\log\left(\frac{\lambda\epsilon_0}{\gamma\epsilon}\right)\right)$ queries to $h_F$.*

## 5 Lower Bounds for Randomized Algorithms

We now turn to randomized algorithms for (1). In the deterministic constructions, we relied on being able to set $v_r$ and $\delta_{i,r}$ based on the predictable behavior of the algorithm. This is impossible for randomized algorithms, we must choose the "hard" function before we know the random choices the algorithm will make—so the function must be "hard" more generally than before.

Previously, we chose vectors $v_r$ orthogonal to all previous queries made by the algorithm. For randomized algorithms this cannot be ensured. However, if we choose orthonormal vectors $v_r$ randomly in a high dimensional space, they will be *nearly* orthogonal to queries with high probability. Slightly modifying the absolute or squared difference from before makes near orthogonality sufficient. This issue increases the required dimension but does not otherwise affect the lower bounds.

More problematic is our inability to anticipate the order in which the algorithm will query the components, precluding the use of $\delta_{i,r}$. In the deterministic setting, if a term revealing a new $v_r$ appeared in half of the components, we could ensure that the algorithm must make $m/2$ queries to find it. However, a randomized algorithm could find it in two queries in expectation, which would eliminate the linear dependence on $m$ in the lower bound! Alternatively, if only one component included the term, a randomized algorithm would indeed need $\Omega(m)$ queries to find it, but that term's effect on suboptimality of $F$ would be scaled down by $m$, again eliminating the dependence on $m$.

To establish a $\Omega(\sqrt{m})$ lower bound for randomized algorithms we must take a new approach. We define $\lfloor \frac{m}{2} \rfloor$ pairs of functions which operate on $\lfloor \frac{m}{2} \rfloor$ orthogonal subspaces of $\mathbb{R}^d$. Each pair of functions resembles the constructions from the previous section, but since there are many of them, the algorithm must solve $\Omega(m)$ separate optimization problems in order to optimize $F$.

### 5.1 Lipschitz Continuous Components

First consider the non-smooth, non-strongly-convex setting and assume for simplicity $m$ is even (otherwise we simply let the last function be zero). We define the helper function $\psi_c$, which replaces the absolute value operation and makes our construction resistant to small inner products between iterates and not-yet-discovered components:

$$\psi_c(z) = \max\left(0, |z| - c\right) \tag{9}$$

Next, we define $m/2$ pairs of functions, indexed by $i = 1..m/2$:

$$f_{i,1}(x) = \frac{1}{\sqrt{2}} |b - \langle x, \, v_{i,0}\rangle| + \frac{1}{2\sqrt{k}} \sum_{r \text{ even}}^{k} \psi_c\left(\langle x, \, v_{i,r-1}\rangle - \langle x, \, v_{i,r}\rangle\right) \tag{10}$$

$$f_{i,2}(x) = \frac{1}{2\sqrt{k}} \sum_{r \text{ odd}}^{k} \psi_c\left(\langle x, \, v_{i,r-1}\rangle - \langle x, \, v_{i,r}\rangle\right)$$

where $\{v_{i,r}\}_{r=0..k, i=1..m/2}$ are random orthonormal vectors and $k = \Theta(\frac{1}{\epsilon\sqrt{m}})$. With $c$ sufficiently small and the dimensionality sufficiently high, with high probability the algorithm only learns the

identity of new vectors $v_{i,r}$ by alternately querying $f_{i,1}$ and $f_{i,2}$; so revealing all $k+1$ vectors requires at least $k+1$ total queries. Until $v_{i,k}$ is revealed, an iterate is $\Omega(\epsilon)$-suboptimal on $(f_{i,1} + f_{i,2})/2$. From here, we show that an $\epsilon$-suboptimal solution to $F(x)$ can be found only after at least $k+1$ queries are made to at least $m/4$ pairs, for a total of $\Omega(mk)$ queries. This time, since the optimum $x^*$ will need to have inner product $b$ with $\Theta(mk)$ vectors $v_{i,r}$, we need to have $b = \Theta(\frac{1}{\sqrt{mk}}) = \Theta(\sqrt{\epsilon}/\sqrt{m})$, and the total number of queries is $\Omega(mk) = \Omega(\frac{\sqrt{m}}{\epsilon})$. The $\Omega(m)$ term of the lower bound follows trivially since we require $\epsilon = \mathcal{O}(1/\sqrt{m})$, (proofs in Appendix C.1):

**Theorem 5.** *For any $L, B > 0$, any $0 < \epsilon < \frac{LB}{10\sqrt{m}}$, any $m \geq 2$, and any randomized algorithm $A$ with access to $h_F$, there exists a dimension $d = \mathcal{O}\left(\frac{L^4 B^6}{\epsilon^4} \log\left(\frac{LB}{\epsilon}\right)\right)$, and $m$ functions $f_i$ defined over $\mathcal{X} = \{x \in \mathbb{R}^d : \|x\| \leq B\}$, which are convex and $L$-Lipschitz continuous, such that to find a point $\hat{x}$ for which $\mathbb{E}\left[F(\hat{x}) - F(x^*)\right] < \epsilon$, $A$ must make $\Omega\left(m + \frac{\sqrt{m}LB}{\epsilon}\right)$ queries to $h_F$.*

An added regularizer gives the result for strongly convex functions (see Appendix C.2):

**Theorem 6.** *For any $L, \lambda > 0$, any $0 < \epsilon < \frac{L^2}{200\lambda m}$, any $m \geq 2$, and any randomized algorithm $A$ with access to $h_F$, there exists a dimension $d = \mathcal{O}\left(\frac{L^4}{\lambda^3 \epsilon} \log \frac{L}{\sqrt{\lambda \epsilon}}\right)$, and $m$ functions $f_i$ defined over $\mathcal{X} \subseteq \mathbb{R}^d$, which are $L$-Lipschitz continuous and $\lambda$-strongly convex, such that in order to find a point $\hat{x}$ for which $\mathbb{E}\left[F(\hat{x}) - F(x^*)\right] < \epsilon$, $A$ must make $\Omega\left(m + \frac{\sqrt{m}L}{\sqrt{\lambda \epsilon}}\right)$ queries to $h_F$.*

The large dimension required by these lower bounds is the cost of omitting the assumption that the algorithm's queries lie in the span of previous oracle responses. If we do assume that the queries lie in that span, the necessary dimension is only on the order of the number of oracle queries needed.

When $\epsilon = \Omega\left(LB/\sqrt{m}\right)$ in the non-strongly convex case or $\epsilon = \Omega\left(L^2/(\lambda m)\right)$ in the strongly convex case, the lower bounds for randomized algorithms presented above do not apply. Instead, we can obtain a lower bound based on an information theoretic argument. We first uniformly randomly choose a parameter $p$, which is either $(1/2 - 2\epsilon)$ or $(1/2 + 2\epsilon)$. Then for $i = 1, ..., m$, in the non-strongly convex case we make $f_i(x) = x$ with probability $p$ and $f_i(x) = -x$ with probability $1 - p$. Optimizing $F(x)$ to within $\epsilon$ accuracy then implies recovering the bias of the Bernoulli random variable, which requires $\Omega(1/\epsilon^2)$ queries based on a standard information theoretic result [2, 19]. Setting $f_i(x) = \pm x + \frac{\lambda}{2} \|x\|^2$ gives a $\Omega(1/(\lambda \epsilon))$ lower bound in the $\lambda$-strongly convex setting. This is formalized in Appendix C.5.

## 5.2 Smooth Components

When the functions $f_i$ are smooth and not strongly convex, we define another helper function $\phi_c$:

$$\phi_c(z) = \begin{cases} 0 & |z| \leq c \\ 2(|z| - c)^2 & c < |z| \leq 2c \\ z^2 - 2c^2 & |z| > 2c \end{cases} \tag{11}$$

and the following pairs of functions for $i = 1, ..., m/2$:

$$f_{i,1}(x) = \frac{1}{16}\left( \langle x, v_{i,0} \rangle^2 - 2a \langle x, v_{i,0} \rangle + \sum_{r \text{ even}}^{k} \phi_c\left(\langle x, v_{i,r-1} \rangle - \langle x, v_{i,r} \rangle\right) \right) \tag{12}$$

$$f_{i,2}(x) = \frac{1}{16}\left( \phi_c\left(\langle x, v_{i,k} \rangle\right) + \sum_{r \text{ odd}}^{k} \phi_c\left(\langle x, v_{i,r-1} \rangle - \langle x, v_{i,r} \rangle\right) \right)$$

with $v_{i,r}$ as before. The same arguments apply, after replacing the absolute difference with squared difference. A separate argument is required in this case for the $\Omega(m)$ term in the bound, which we show using a construction involving $m$ simple linear functions (see Appendix C.3).

**Theorem 7.** *For any $\gamma, B, \epsilon > 0$, any $m \geq 2$, and any randomized algorithm $A$ with access to $h_F$, there exists a sufficiently large dimension $d = \mathcal{O}\left(\frac{\gamma^2 B^6}{\epsilon^2} \log\left(\frac{\gamma B^2}{\epsilon}\right) + B^2 m \log m\right)$ and $m$ functions $f_i$ defined over $\mathcal{X} = \{x \in \mathbb{R}^d : \|x\| \leq B\}$, which are convex and $\gamma$-smooth, such that to find a point $\hat{x} \in \mathbb{R}^d$ for which $\mathbb{E}\left[F(\hat{x}) - F(x^*)\right] < \epsilon$, $A$ must make $\Omega\left(m + \sqrt{\frac{m\gamma B^2}{\epsilon}}\right)$ queries to $h_F$.*

In the strongly convex case, we add the term $\lambda \left\| x \right\|^2 / 2$ to $f_{i,1}$ and $f_{i,2}$ (see Appendix C.4) to obtain:

**Theorem 8.** *For any $m \geq 2$, any $\gamma, \lambda > 0$ such that $\frac{\gamma}{\lambda} > 161m$, any $\epsilon > 0$, any $\epsilon_0 > 60\epsilon\sqrt{\frac{\gamma}{\lambda m}}$, and any randomized algorithm $A$, there exists a dimension $d = \mathcal{O}\left( \frac{\gamma^{2.5}\epsilon_0}{\lambda^{2.5}\epsilon} \log^3\left( \frac{\lambda\epsilon_0}{\gamma\epsilon} \right) + \frac{m\gamma\epsilon_0}{\lambda\epsilon} \log m \right)$, domain $\mathcal{X} \subseteq \mathbb{R}^d$, $x_0 \in \mathcal{X}$, and $m$ functions $f_i$ defined on $\mathcal{X}$ which are $\gamma$-smooth and $\lambda$-strongly convex, and such that $F(x_0) - F(x^*) = \epsilon_0$ and such that in order to find a point $\hat{x} \in \mathcal{X}$ such that $\mathbb{E}\left[ F(\hat{x}) - F(x^*) \right] < \epsilon$, $A$ must make $\Omega\left( m + \sqrt{\frac{m\gamma}{\lambda}} \log\left( \frac{\epsilon_0}{\epsilon} \sqrt{\frac{m\lambda}{\gamma}} \right) \right)$ queries to $h_F$.*

**Remark:** We consider (1) as a constrained optimization problem, thus the minimizer of $F$ could be achieved on the boundary of $\mathcal{X}$, meaning that the gradient need not vanish. If we make the additional assumption that the minimizer of $F$ lies on the *interior* of $\mathcal{X}$ (and is thus the unconstrained global minimum), Theorems 1-8 all still apply, with a slight modification to Theorems 3 and 7. Since the gradient now needs to vanish on $\mathcal{X}$, 0 is always $\tilde{\mathcal{O}}(\gamma B^2)$-suboptimal, and only values of $\epsilon$ in the range $0 < \epsilon < \frac{\gamma B^2}{128}$ and $0 < \epsilon < \frac{9\gamma B^2}{128}$ result in a non-trivial lower bound (see Remarks at the end of Appendices B.3 and C.3).

# 6 Conclusion

We provide a tight (up to a log factor) understanding of optimizing finite sum problems of the form (1) using a component prox oracle.

Randomized optimization of (1) has been the subject of much research in the past several years, starting with the presentation of SDCA and SAG, and continuing with accelerated variants. Obtaining lower bounds can be very useful for better understanding the problem, for knowing where it might or might not be possible to improve or where different assumptions would be needed to improve, and for establishing optimality of optimization methods. Indeed, several attempts have been made at lower bounds for the finite sum setting [1, 9]. But as we explain in the introduction, these were unsatisfactory and covered only limited classes of methods. Here we show that in a fairly general sense, accelerated SDCA, SVRG, SAG, and KATYUSHA are optimal up to a log factor. Improving on their runtime would require additional assumptions, or perhaps a stronger oracle. However, even if given "full" access to the component functions, all algorithms that we can think of utilize this information to calculate a prox vector. Thus, it is unclear what realistic oracle would be more powerful than the prox oracle we consider.

Our results highlight the power of randomization, showing that no deterministic algorithm can beat the linear dependence on $m$ and reduce it to the $\sqrt{m}$ dependence of the randomized algorithms.

The deterministic algorithm for non-smooth problems that we present in Section 3 is also of interest in its own right. It avoids randomization, which is not usually problematic, but makes it fully parallelizable unlike the optimal stochastic methods. Consider, for example, a supervised learning problem where $f_i(x) = \ell(\langle \phi_i, x \rangle, y_i)$ is the (non-smooth) loss on a single training example $(\phi_i, y_i)$, and the data is distributed across machines. Calculating a prox oracle involves applying the Fenchel conjugate of the loss function $\ell$, but even if a closed form is not available, this is often easy to compute numerically, and is used in algorithms such as SDCA. But unlike SDCA, which is inherently sequential, we can calculate all $m$ prox operations in parallel on the different machines, average the resulting gradients of the smoothed function, and take an accelerated gradient step to implement our optimal deterministic algorithm. This method attains a recent lower bound for distributed optimization, resolving a question raised by Arjevani and Shamir [5], and when the number of machines is very large improves over all other known distributed optimization methods for the problem.

In studying finite sum problems, we were forced to explicitly study lower bounds for randomized optimization as opposed to stochastic optimization (where the source of randomness is the oracle, not the algorithm). Even for the classic problem of minimizing a smooth function using a first order oracle, we could not locate a published proof that applies to randomized algorithms. We provide a simple construction using $\epsilon$-insensitive differences that allows us to easily obtain such lower bounds without reverting to assuming the iterates are spanned by previous responses (as was done, e.g., in [9]), and could potentially be useful for establishing randomized lower bounds also in other settings.

**Acknowledgements:** We thank Ohad Shamir for his helpful discussions and for pointing out [4].

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
