[Supplementary Material]

# A  Upper bounds for non-smooth sums

Consider the case where the components are not strongly convex. As shown in lemma 1, we can use a single call to a prox oracle to obtain the gradient of

$$f^{(\beta)}(x) = \inf_{u \in \mathcal{X}} f(u) + \frac{\beta}{2} \|x - u\|^2$$

which is a $\beta$-smooth approximation to $f$. We then consider the new optimization problem:

$$\min_{x \in \mathcal{X}} \left\{ \tilde{F}^{(\beta)}(x) := \frac{1}{m} \sum_{i=1}^{m} f_i^{(\beta)}(x) \right\}. \tag{13}$$

Also by lemma 1, setting $\beta = \frac{L^2}{\epsilon}$ ensures that $\tilde{F}^{(\beta)}(x) \leq F(x) \leq \tilde{F}^{(\beta)}(x) + \frac{\epsilon}{2}$ for all $x$. Consequently, any point which is $\frac{\epsilon}{2}$-suboptimal for $\tilde{F}^{(\beta)}$ will be $\epsilon$-suboptimal for $F$. This technique therefore reduces the task of optimizing an instance of an $L$-Lipschitz finite sum to that of optimizing an $\frac{L^2}{\epsilon}$-smooth finite sum.

Solving (13) to $\frac{\epsilon}{2}$-suboptimality using AGD requires $\mathcal{O}\left(\frac{mLB}{\epsilon}\right)$ gradients for $\tilde{F}^{(\beta)}$ which requires that same number of prox oracles from $h_F$. Formally:

**Theorem 9.** *For any $L, B > 0$, any $\epsilon < LB$, and any $m \geq 1$ functions $f_i$ which are convex and $L$-Lipschitz continuous over the domain $\mathcal{X} \subseteq \{x \in \mathbb{R}^d : \|x\| \leq B\}$, applying AGD to (13) for $\beta = \frac{L^2}{\epsilon}$, will result in a point $\hat{x}$ such that $F(\hat{x}) - F(x^*) < \epsilon$ after $\mathcal{O}\left(\frac{mLB}{\epsilon}\right)$ queries to $h_F$.*

When the component functions are $\lambda$-strongly convex, a more sophisticated strategy is required to avoid an extra log factor. The solution is the AdaptSmooth algorithm [4]. This involves solving $\mathcal{O}(\log \frac{1}{\epsilon})$ smooth and strongly convex subproblems, where the $t^{\text{th}}$ subproblem is reducing the suboptimality of the $\beta_t$-smooth and $\lambda$-strongly convex function $F^{(\beta_t)}(x)$ by a factor of four, where $\beta_t = \frac{L^2}{\epsilon_0} 2^t$ and where $\epsilon_0 \leq \frac{L^2}{\lambda}$ upper bounds the initial suboptimality. Using this method results in an $\epsilon$-suboptimal solution for $F$ after $\sum_{t=0}^{\log \frac{\epsilon_0}{\epsilon}} \text{Time}(\beta_t, \lambda)$ queries to $h_F$.

In the case of AGD, $\text{Time}(\gamma, \lambda) = \mathcal{O}\left(m\sqrt{\frac{\gamma}{\lambda}}\right)$ and

$$\sum_{t=0}^{\log \frac{\epsilon_0}{\epsilon}} \text{Time}\left(\frac{L^2}{\epsilon_0} 2^t, \lambda\right) = \mathcal{O}\left(\frac{mL}{\sqrt{\lambda \epsilon}}\right)$$

**Theorem 10.** *For any $L, \lambda, \epsilon > 0$, and any $m \geq 1$ functions $f_i$, which are $L$-Lipschitz continuous and $\lambda$-strongly convex on the domain $\mathcal{X} \subseteq \mathbb{R}^d$, applying AdaptSmooth with AGD will find a point $\hat{x} \in \mathcal{X}$ such that $F(\hat{x}) - F(x^*) < \epsilon$ after $\mathcal{O}\left(\frac{mL}{\sqrt{\lambda \epsilon}}\right)$ queries to $h_F$.*

To conclude our presentation of upper bounds, we emphasize that the smoothing methods described in this section will only improve oracle complexity when used with accelerated methods. For example, using non-accelerated gradient descent on $\tilde{F}^{(\beta)}$ in the not strongly convex case leads to an oracle complexity of $\mathcal{O}\left(\frac{mL^2B^2}{\epsilon^2}\right)$, which is no better than the convergence rate of gradient descent applied directly to $F$.

# B  Lower bounds for deterministic algorithms

## B.1  Non-smooth and not strongly convex components

**Theorem 1.** *For any $L, B > 0$, any $0 < \epsilon < \frac{LB}{12}$, any $m \geq 2$, and any deterministic algorithm $A$ with access to $h_F$, there exists a dimension $d = \mathcal{O}\left(\frac{mLB}{\epsilon}\right)$, and $m$ functions $f_i$ defined over $\mathcal{X} = \{x \in \mathbb{R}^d : \|x\| \leq B\}$, which are convex and $L$-Lipschitz continuous, such that in order to find a point $\hat{x}$ for which $F(\hat{x}) - F(x^*) < \epsilon$, $A$ must make $\Omega\left(\frac{mLB}{\epsilon}\right)$ queries to $h_F$.*

*Proof.* Without loss of generality, we can assume $L = B = 1$. For particular values $b$ and $k$ to be decided upon later, we use the functions (6):

$$f_i(x) = \frac{1}{\sqrt{2}} |b - \langle x, v_0 \rangle| + \frac{1}{2\sqrt{k}} \sum_{r=1}^{k} \delta_{i,r} |\langle x, v_{r-1} \rangle - \langle x, v_r \rangle|$$

It is straightforward to confirm that $f_i$ is both 1-Lipschitz and convex (for orthonormal vectors $v_r$ and indicators $\delta_{i,r} \in \{0,1\}$). As explained in the main text, the orthonormal vectors $v_r \in \mathbb{R}^d$ and indicators $\delta_{i,r} \in \{0,1\}$ are chosen according to the behavior of the algorithm $A$. At the end of each round $t$, we set $\delta_{i,t} = 1$ iff the algorithm did *not* query function $i$ during round $t$ (and zero otherwise), and we set $v_t$ to be orthogonal to the vectors $\{v_0, ..., v_{t-1}\}$ as well as every query made by the algorithm so far. Orthogonalizing the vectors in this way is possible as long as the dimension is at least as large as the number of oracle queries $A$ has made so far plus $t$. We are allowed to construct $v_t$ and $\delta_{i,t}$ in this way as long as the algorithm's execution up until round $t$, and thus our choice of $v_t$ and $\delta_{i,t}$, depends only on $v_r$ and $\delta_{i,r}$ for $r < t$. We can enforce this condition by answering the queries during round $t$ according to

$$f_i^t(x) = \frac{1}{\sqrt{2}} |b - \langle x, v_0 \rangle| + \frac{1}{2\sqrt{k}} \sum_{r=1}^{t-1} \delta_{i,r} |\langle x, v_{r-1} \rangle - \langle x, v_r \rangle|$$

For non-smooth functions, the subgradient oracle is not uniquely defined—-many different subgradients might be a valid response. However, in order to say that an algorithm successfully optimizes a function, it must be able to do so no matter which subgradient is receives. Conversely, to show a lower bound, it is sufficient to show that for *some* valid subgradient the algorithm fails. And so, in constructing a "hard" instance to optimize we are actually constructing both a function and a subgradient oracle for it, with specific subgradient responses. Therefore, answering the algorithm's queries during round $t$ according to $f_i^t$ is valid so long as the subgradient we return is a valid subgradient for $f_i$ (the converse need not be true) and the prox returned is exactly the prox of $f_i$. For now, assume that this query-answering strategy is consistent (we will prove this last).

Then if $d = \lceil \frac{m}{\epsilon} \rceil + k + 1$ and if $x$ is an iterate generated both before $A$ completes round $k$ and before it makes $\lceil \frac{m}{\epsilon} \rceil$ queries to $h_F$ (so that the dimension is large enough to orthogonalize each $v_t$ as described above), then $\langle x, v_k \rangle = 0$ by construction. This allows us to bound the suboptimality of $F(x)$ (since $\lceil \frac{m}{2} \rceil$ functions are queried during each round, $\sum_{i=1}^{m} \delta_{i,r} = \lfloor \frac{m}{2} \rfloor$):

$$F(x) = \frac{1}{m} \sum_{i=1}^{m} f_i(x)$$

$$= \frac{1}{\sqrt{2}} |b - \langle x, v_0 \rangle| + \frac{\lfloor \frac{m}{2} \rfloor}{2m\sqrt{k}} \sum_{r=1}^{k} |\langle x, v_{r-1} \rangle - \langle x, v_r \rangle|$$

$F$ is non-negative and $F(x_b) = 0$ where $x_b = b \sum_{r=0}^{k} v_r$. Choosing $b = \frac{1}{\sqrt{k+1}}$ makes $\|x_b\| = 1$ so that $x_b \in \mathcal{X}$. Therefore, $F$ achieves its minimum on $\mathcal{X}$ and

$$F(x) - F(x^*) = \frac{1}{\sqrt{2}} |b - \langle x, v_0 \rangle| + \frac{\lfloor \frac{m}{2} \rfloor}{2m\sqrt{k}} \sum_{r=1}^{k} |\langle x, v_{r-1} \rangle - \langle x, v_r \rangle| - 0$$

$$\geq \frac{1}{\sqrt{2}} |b - \langle x, v_0 \rangle| + \frac{1}{6\sqrt{k}} |\langle x, v_0 \rangle - \langle x, v_k \rangle|$$

$$= \frac{1}{\sqrt{2}} |b - \langle x, v_0 \rangle| + \frac{1}{6\sqrt{k}} |\langle x, v_0 \rangle|$$

$$\geq \min_{z \in \mathbb{R}} \frac{1}{\sqrt{2}} |b - z| + \frac{1}{6\sqrt{k}} |z|$$

$$= \frac{b}{6\sqrt{k}}$$

$$\geq \frac{1}{12k}$$

Where the final inequality holds when $k \geq 1$. Setting $k = \lfloor \frac{1}{12\epsilon} \rfloor$ implies $F(x) - F(x^*) \geq \epsilon$. Therefore, $A$ must either query $h_F$ more than $\lceil \frac{m}{\epsilon} \rceil$ times or complete $k$ rounds to reach an $\epsilon$-suboptimal solution. Completing each round requires at least $\lceil \frac{m}{2} \rceil$ queries to $h_F$, so when $\epsilon \leq \frac{1}{12}$, this implies a lower bound of

$$\min \left( \frac{m}{\epsilon}, \left\lfloor \frac{1}{12\epsilon} \right\rfloor \frac{m}{2} \right) \geq \frac{m}{48\epsilon}$$

To complete the proof, it remains to show that the subgradients and proxs of $f_i^t$ are consistent with those of $f_i$ at every time $t$. Since every function operates on the $(k+1)$-dimensional subspace of $\mathbb{R}^d$ spanned by $\{v_r\}$, it will be convenient to decompose vectors into two components: $x = x^v + x^\perp$ where $x^v = \sum_{r=0}^{k} \langle x, v_r \rangle v_r$ and $x^\perp = x - x^v$. Note that $f_i^t(x) = f_i^t(x^v)$.

**Lemma 2.** *For any $t \leq k$ and any $x$ such that $x^v \in span\{v_0, v_1, ..., v_{t-1}\}$, if function $i$ is queried during round $t$, then $\partial f_i^t(x) \subseteq \partial f_i(x)$.*

*Proof.* All subgradients of $f_i$ have the form

$$\frac{\text{sign}(b - \langle x, v_0 \rangle)}{\sqrt{2}} v_0 + \frac{1}{2\sqrt{k}} \sum_{r=1}^{k} \delta_{i,r} \text{sign}(\langle x, v_{r-1} \rangle - \langle x, v_r \rangle)(v_{r-1} - v_r)$$

where we define $\text{sign}(0) = 0$. Since function $i$ is queried during round $t$, $\delta_{i,t} = 0$, and since $\langle x, v_{r-1} \rangle = 0 = \langle x, v_r \rangle$ for all $r > t$, $\partial f_i(x)$ contains all subgradients of the form

$$\frac{\text{sign}(b - \langle x, v_0 \rangle)}{\sqrt{2}} v_0 + \frac{1}{2\sqrt{k}} \sum_{r=1}^{t-1} \delta_{i,r} \text{sign}(\langle x, v_{r-1} \rangle - \langle x, v_r \rangle)(v_{r-1} - v_r)$$

which is exactly $\partial f_i^t(x)$. □

**Lemma 3.** *For any $t \leq k$ and any $x$ such that $x^v \in span\{v_0, v_1, ..., v_{t-1}\}$, if function $i$ is queried during round $t$ then $\forall \beta > 0$, $prox_{f_i^t}(x, \beta) = prox_{f_i}(x, \beta)$.*

*Proof.* Consider the definition of the prox oracle from equation 3

$$\text{prox}_{f_i}(x, \beta) = \arg\min_u f_i(u) + \frac{\beta}{2} \|x - u\|^2$$

$$= \arg\min_{u^v, u^\perp} f_i(u^v) + \frac{\beta}{2} \|x^v + x^\perp - u^v - u^\perp\|^2$$

$$= \arg\min_{u^v} f_i(u^v) + \frac{\beta}{2} \|x^v - u^v\|^2 + \arg\min_{u^\perp} \frac{\beta}{2} \|x^\perp - u^\perp\|^2$$

$$= x^\perp + \text{prox}_{f_i}(x^v, \beta)$$

Next, we further decompose $x^v = x^- + x^+$ where

$$x^- = \sum_{r=0}^{t-1} \langle x^v, v_r \rangle v_r \qquad \text{and} \qquad x^+ = \sum_{r=t}^{k} \langle x^v, v_r \rangle v_r$$

Note that $x^+ = 0$ and since function $i$ is queried during round $t$, $\delta_{i,t} = 0$. Therefore,

$$\text{prox}_{f_i}(x^v, \beta) = \arg\min_{u^-, u^+} f_i(u^- + u^+) + \frac{\beta}{2} \|x^- - u^- - u^+\|^2 \tag{14}$$

$$= \arg\min_{u^-, u^+} \frac{1}{\sqrt{2}} |b - \langle u^-, v_0 \rangle| + \frac{1}{2\sqrt{k}} \sum_{r=1}^{t-1} \delta_{i,r} |\langle u^-, v_{r-1} \rangle - \langle u^-, v_r \rangle|$$

$$+ \frac{1}{2\sqrt{k}} \sum_{r=t+1}^{k} \delta_{i,r} |\langle u^+, v_{r-1} \rangle - \langle u^+, v_r \rangle| + \frac{\beta}{2} \left( \|x^- - u^-\|^2 + \|u^+\|^2 \right)$$

$$= \text{prox}_{f_i^t}(x^v, \beta)$$

The last equality follows from the fact that that the minimization is completely separable between $u^-$ and $u^+$, allowing us to minimized over each variable separately. The terms containing $u^+$ are non-negative and can be simultaneously equal to 0 when $u^+ = 0$. Therefore, $\text{prox}_{f_i^t}(x, \beta) = \text{prox}_{f_i}(x, \beta)$. □

These lemmas show that the subgradients and proxs of $f_i^t$ at vectors which are queried during round $t$ are consistent with the subgradients and proxs of $f_i$. This confirms that our construction is sound, and completes the proof. □

## B.2 Non-smooth and strongly convex components

**Theorem 2.** *For any $L, \lambda > 0$, any $0 < \epsilon < \frac{L^2}{288\lambda}$, any $m \geq 2$, and any deterministic algorithm $A$ with access to $h_F$, there exists a dimension $d = \mathcal{O}\left(\frac{mL}{\sqrt{\lambda\epsilon}}\right)$, and $m$ functions $f_i$ defined over $\mathcal{X} \subseteq \mathbb{R}^d$, which are $L$-Lipschitz continuous and $\lambda$-strongly convex, such that in order to find a point $\hat{x}$ for which $F(\hat{x}) - F(x^*) < \epsilon$, $A$ must make $\Omega\left(\frac{mL}{\sqrt{\lambda\epsilon}}\right)$ queries to $h_F$.*

*Proof.* Suppose towards contradiction that the contrary were true, and there is an $A$ which can find a point $\hat{x}$ for which $F(\hat{x}) - F(x^*) < \epsilon$ after at most $o\left(\frac{mL}{\sqrt{\lambda\epsilon}}\right)$ queries to $h_F$. Then $A$ could be used to minimize the sum $\tilde{F}$ of $m$ functions $\tilde{f}_i$, which are convex and $L$-Lipschitz continuous over a domain of $\{x : \|x\| \leq B\}$ by adding a regularizer. Let

$$F(x) = \frac{1}{m} \sum_{i=1}^{m} f_i(x) := \frac{1}{m} \sum_{i=1}^{m} \tilde{f}_i(x) + \frac{\lambda}{2} \|x\|^2$$

Note that $f_i$ is $\lambda$-strongly convex and since $\tilde{f}_i$ is $L$-Lipschitz on the $B$-bounded domain, $f_i$ is $(L + \lambda B)$-Lipschitz continuous on the same domain. Furthermore, by setting $\lambda = \frac{\epsilon}{B^2}$,

$$\tilde{F}(x) \leq F(x) \leq \tilde{F}(x) + \frac{\epsilon}{2B^2} \|x\|^2 \leq \tilde{F}(x) + \frac{\epsilon}{2}$$

By assumption, $A$ can find an $\hat{x}$ such that $F(\hat{x}) - F(x^*) < \frac{\epsilon}{2}$ using $o\left(\frac{m(L+\lambda B)}{\sqrt{\lambda\epsilon}}\right) = o\left(\frac{mLB}{\epsilon}\right)$ queries to $h_F$, and

$$\frac{\epsilon}{2} > F(\hat{x}) - F(x^*) \geq \tilde{F}(\hat{x}) - \tilde{F}(\tilde{x}^*) - \frac{\epsilon}{2}$$

Thus $\hat{x}$ is $\epsilon$-suboptimal for $\tilde{F}$. However, this contradicts the conclusion of theorem 1 when the parameters of the strongly convex problem correspond to parameters of a non-strongly convex problem to which theorem 1 applies. In particular, for any values $L > 0$, $\lambda > 0$, $0 < \epsilon < \frac{L^2}{288\lambda}$, and dimension $d = \mathcal{O}\left(\frac{mL}{\sqrt{\lambda\epsilon}}\right)$ there is a contradiction. $\qquad\square$

## B.3 Smooth and not strongly convex components

**Theorem 3.** *For any $\gamma, B, \epsilon > 0$, any $m \geq 2$, and any deterministic algorithm $A$ with access to $h_F$, there exists a sufficiently large dimension $d = \mathcal{O}\left(m\sqrt{\gamma B^2/\epsilon}\right)$, and $m$ functions $f_i$ defined over $\mathcal{X} = \{x \in \mathbb{R}^d : \|x\| \leq B\}$, which are convex and $\gamma$-smooth, such that in order to find a point $\hat{x} \in \mathbb{R}^d$ for which $F(\hat{x}) - F(x^*) < \epsilon$, $A$ must make $\Omega\left(m\sqrt{\gamma B^2/\epsilon}\right)$ queries to $h_F$.*

*Proof.* This proof will be very similar to the proof of theorem 1. Without loss of generality, we can assume that $\gamma = B = 1$. For a values $a$ and $k$ to be fixed later, we define:

$$f_i(x) = \frac{1}{8}\left(\delta_{i,1}\left(\langle x, v_0\rangle^2 - 2a\langle x, v_0\rangle\right) + \sum_{r=1}^{k} \delta_{i,r}\left(\langle x, v_{r-1}\rangle - \langle x, v_r\rangle\right)^2 + \delta_{i,k}\langle x, v_k\rangle^2\right)$$

We define the orthonormal vectors $v_r \in \mathbb{R}^d$ and indicators $\delta_{i,t} \in \{0, 1\}$ as in the proof of theorem 1. That is, at the end of round $t$, we set $\delta_{i,t} = 1$ if the algorithm $A$ does *not* query function $i$ during round $t$ (and zero otherwise) and we construct $v_t$ to be orthogonal to $\{v_0, ..., v_{t-1}\}$ as well as every point queried by the algorithm so far. Orthogonalizing the vectors is possible as long as the dimension is at least as large as the number of oracle queries $A$ has made so far plus $t$. As before, we are allowed to construct $v_t$ and $\delta_{i,t}$ in this way as long as the algorithm's execution up until round $t$, and thus our choice of $v_t$ and $\delta_{i,t}$, depends only on $v_r$ and $\delta_{i,r}$ for $r < t$. We enforce this condition by answering the queries during round $t < k$ according to

$$f_i^t(x) = \frac{1}{8}\left(\delta_{i,1}\left(\langle x, v_0\rangle^2 - 2a\langle x, v_0\rangle\right) + \sum_{r=1}^{t-1} \delta_{i,r}\left(\langle x, v_{r-1}\rangle - \langle x, v_r\rangle\right)^2\right)$$

We will assume for now that this query-answering strategy is self-consistent, and prove it later. This allows us to bound the suboptimality of $F(x)$. Note that since exactly $\lceil \frac{m}{2} \rceil$ functions are queried each round, $\sum_{i=1}^{m} \delta_{i,r} = \lfloor \frac{m}{2} \rfloor$, so let

$$
\begin{aligned}
F^t(x) &= \frac{1}{m} \sum_{i=1}^{m} f_i^t(x) + \delta_{i,t} \langle x,\, v_{t-1} \rangle^2 \\
&= \frac{\lfloor \frac{m}{2} \rfloor}{8m} \left( \langle x,\, v_0 \rangle^2 - 2a \langle x,\, v_0 \rangle + \sum_{r=1}^{t-1} (\langle x,\, v_{r-1} \rangle - \langle x,\, v_r \rangle)^2 + \langle x,\, v_{t-1} \rangle^2 \right)
\end{aligned}
$$

Then if $d = \lceil \frac{m}{\sqrt{\epsilon}} \rceil + k + 1$, and if $x$ is an iterate generated both before $A$ completes round $q := \lfloor \frac{k}{2} \rfloor$ and before it makes $\lceil \frac{m}{\sqrt{\epsilon}} \rceil$ queries to $h_F$, then $\langle x,\, v_r \rangle = 0$ for all $r \geq q$ by construction. Then, for this $x$, $F^q(x) = F^{k+1}(x) = F(x)$. By first order optimality conditions for $F^t$, its optimum $x_t^*$ must satisfy that:

$$
\begin{aligned}
2 \langle x_t^*,\, v_0 \rangle - \langle x_t^*,\, v_1 \rangle &= a \\
\langle x_t^*,\, v_{r-1} \rangle - 2 \langle x_t^*,\, v_r \rangle + \langle x_t^*,\, v_{r+1} \rangle &= 0 \quad \text{for } 1 \leq r \leq t - 2 \\
\langle x_t^*,\, v_{t-2} \rangle - 2 \langle x_t^*,\, v_{t-1} \rangle &= 0
\end{aligned}
$$

It is straightforward to confirm that the solution to this system of equations is

$$
x_t^* = a \sum_{r=0}^{t-1} \left( 1 - \frac{r+1}{t+1} \right) v_r
$$

that

$$
F^t(x_t^*) = -\frac{a^2 \lfloor \frac{m}{2} \rfloor}{8m} \left( 1 - \frac{1}{t+1} \right)
$$

and that

$$
\begin{aligned}
\|x_t^*\|^2 &= a^2 \sum_{r=0}^{t-1} \left( 1 - \frac{r+1}{t+1} \right)^2 \\
&= a^2 \left( t - \frac{2}{t+1} \sum_{r=0}^{t-1} (r+1) + \frac{1}{(t+1)^2} \sum_{r=0}^{t-1} (r+1)^2 \right) \\
&= a^2 \left( t - \frac{2}{t+1} \frac{t(t+1)}{2} + \frac{1}{(t+1)^2} \frac{t(t+1)(2t+1)}{6} \right) \\
&\leq \frac{a^2 t}{3}
\end{aligned}
$$

Thus, we set $a = \sqrt{\frac{3}{k+1}}$, ensuring $\|x_{k+1}^*\| = 1$ so that $x_{k+1}^* = x^* \in \mathcal{X}$. Furthermore, for the iterate $x$ made before $q$ rounds of queries,

$$
\begin{aligned}
F(x) - F(x^*) &= F^q(x) - F^{k+1}(x_{k+1}^*) \\
&\geq F^q(x_q^*) - F^{k+1}(x_{k+1}^*) \\
&= -\frac{3 \lfloor \frac{m}{2} \rfloor}{8m(k+1)} \left( 1 - \frac{1}{\lfloor \frac{k}{2} \rfloor + 1} \right) + \frac{3 \lfloor \frac{m}{2} \rfloor}{8m(k+1)} \left( 1 - \frac{1}{k+2} \right) \\
&\geq \frac{1}{32 k^2}
\end{aligned}
$$

where the last inequality holds as long as $k \geq 2$. So, when $\epsilon < \frac{1}{128}$ and we let $k = \lfloor \frac{1}{\sqrt{32\epsilon}} \rfloor$, this ensures that

$$
F(x) - F(x^*) = F^q(x) - F^{k+1}(x_{k+1}^*) \geq \epsilon
$$

and therefore, $A$ must complete at least $q$ rounds or make more than $\lceil \frac{m}{\sqrt{\epsilon}} \rceil$ queries to $h_F$ in order to reach an $\epsilon$-suboptimal point. This implies a lower bound of

$$\min\left(\left\lceil \frac{m}{\sqrt{\epsilon}} \right\rceil, \; q\left\lceil \frac{m}{2} \right\rceil\right) \geq \frac{m}{16\sqrt{6}\epsilon}$$

To complete the proof, it remains to show that the gradient and prox of $f_i^t$ is consistent with those of $f_i$ at every time $t$. Since every function operates on the $(k+1)$-dimensional subspace of $\mathbb{R}^d$ spanned by $\{v_r\}$, it will be convenient to decompose vectors into two components: $x = x^v + x^\perp$ where $x^v = \sum_{r=0}^{k} \langle x, v_r \rangle v_r$ and $x^\perp = x - x^v$. Note that $f_i^t(x) = f_i^t(x^v)$.

**Lemma 4.** *For any $t \leq k$ and any $x$ such that $x^v \in span\,\{v_0, v_1, ..., v_{t-1}\}$, if function $i$ is queried during round $t$, then $\nabla f_i^t(x) = \nabla f_i(x)$.*

*Proof.* Since function $i$ is queried during round $t$, $\delta_{i,t} = 0$ so

$$\nabla f_i(x) = \frac{1}{4}\left(\delta_{i,1}\left(\langle x, v_1 \rangle v_0 - 2v_0\right) + \sum_{r=1}^{t-1} \delta_{i,r}\left(\langle x, v_{r-1} \rangle - \langle x, v_r \rangle\right)\left(v_{r-1} - v_r\right)\right) = \nabla f_i^t(x)$$

$\square$

**Lemma 5.** *For any $t \leq k$ and any $x$ such that $x^v \in span\,\{v_0, v_1, ..., v_{t-1}\}$, if function $i$ is queried during round $t$ then $\forall \beta > 0$, $prox_{f_i^t}(x, \beta) = prox_{f_i}(x, \beta)$.*

*Proof.* Up until the last step, this proof is identical to the proof of lemma 3, thus we pick up at (14):

$$\text{prox}_{f_i}(x^v, \beta) = \underset{u^-,u^+}{\arg\min} \; f_i(u^- + u^+) + \frac{\beta}{2}\left\|x^- - u^- - u^+\right\|^2$$

$$= \underset{u^-,u^+}{\arg\min} \frac{1}{8}\left(\delta_{i,1}\left(\langle u^-, v_0 \rangle^2 - 2a\langle u^-, v_0 \rangle\right) + \sum_{r=1}^{t-1} \delta_{i,r}\left(\langle u^-, v_{r-1} - v_r \rangle\right)^2\right.$$

$$+ \sum_{r=t+1}^{k} \delta_{i,r}\left(\langle u^+, v_{r-1} - v_r \rangle\right)^2 + \delta_{i,k}\langle u^+, v_k \rangle^2\right) + \frac{\beta}{2}\left(\left\|x^- - u^-\right\|^2 + \left\|u^+\right\|^2\right)$$

$$= \text{prox}_{f_i^t}(x^v, \beta)$$

The final step comes from the fact that the $\arg\min$ is separable over $u^-$ and $u^+$, meaning we can minimize the two terms individually. The terms which contain $u^+$ are non-negative and equal to zero when $u^+ = 0$. $\square$

These lemmas show that the gradient and prox of $f_i^t$ at vectors which are queried during round $t$ are consistent with the gradient and prox of $f_i$. This confirms that our construction is sound.

This proves the lower bound for $\epsilon < \frac{\gamma B^2}{128}$, we can extend the same lower bound to $\epsilon \geq \frac{\gamma B^2}{128}$ using the following, very simple construction. Let

$$f_i(x) = \begin{cases} 0 & \text{if function } i \text{ is queried in the first } m-1 \text{ queries} \\ 2m\epsilon\,\langle x, v \rangle \end{cases}$$

where $v$ is a unit vector that is orthogonal to all of the first $m-1$ queries. This function is trivially 1-smooth. By construction, the algorithm must make at least $m$ queries to learn the identity of $v$. Until it has done so, any iterate will have objective value zero, while the optimum $F(x^*) = F(v) = -2\epsilon$. Therefore, the algorithm must make at least $m$ queries to reach an $\epsilon$-suboptimal solution. For $\epsilon \geq \frac{\gamma B^2}{128}$

$$m \geq m\sqrt{\frac{\gamma B^2}{128\epsilon}}$$

Therefore a lower bound of $\Omega\left(m\sqrt{\frac{\gamma B^2}{\epsilon}}\right)$ applies for any $\epsilon > 0$. $\square$

**Remark:** If make the additional assumption that $F$ is minimized on the interior of $\mathcal{X}$, since $0$ is $\mathcal{O}(\gamma B^2)$-suboptimal, only $0 < \epsilon < \frac{\gamma B^2}{128}$ gives a non-trivial lower bound. This lower bound is shown by the first construction presented in the previous proof.

## B.4 Smooth and strongly convex components

**Theorem 4.** *For any $\gamma, \lambda > 0$ such that $\frac{\gamma}{\lambda} > 73$, any $\epsilon > 0$, any $\epsilon_0 > \frac{3\gamma\epsilon}{\lambda}$, any $m \geq 2$, and any deterministic algorithm $A$ with access to $h_F$, there exists a sufficiently large dimension $d = \mathcal{O}\left(m\sqrt{\frac{\gamma}{\lambda}}\log\left(\frac{\lambda\epsilon_0}{\gamma\epsilon}\right)\right)$, and $m$ functions $f_i$ defined over $\mathcal{X} \subseteq \mathbb{R}^d$, which are $\gamma$-smooth and $\lambda$-strongly convex and where $F(0) - F(x^*) = \epsilon_0$, such that in order to find a point $\hat{x}$ for which $F(\hat{x}) - F(x^*) < \epsilon$, $A$ must make $\Omega\left(m\sqrt{\frac{\gamma}{\lambda}}\log\left(\frac{\lambda\epsilon_0}{\gamma\epsilon}\right)\right)$ queries to $h_F$.*

*Proof.* We will prove the theorem for 1-smooth and $\lambda$-strongly convex components for any $\lambda < \frac{1}{73}$. This can be extended to arbitrary constants $\gamma$ and $\lambda'$ by taking $\lambda = \frac{\lambda'}{\gamma}$.

For any $k$ and for $\zeta$ and $C$ to be defined later, let

$$f_i(x) = \frac{1-\lambda}{8}\left(\delta_{i,1}\left(\langle x, v_0\rangle^2 - 2C\langle x, v_0\rangle\right) + \delta_{i,k}\zeta\langle x, v_k\rangle^2\right.$$

$$\left. + \sum_{r=1}^{k}\delta_{i,r}\langle x, v_{r-1} - v_r\rangle^2\right) + \frac{\lambda}{2}\|x\|^2$$

where the vectors $v_r$ and indicators $\delta_{i,r}$ are defined in the same way as in the previous proof. This function is just a multiple of the construction in the proof of theorem 3 plus the $\lambda\|x\|^2/2$ term. It is clear that the norm term is uninformative for learning the identity of vectors $v_r$, as the component of the gradients and proxs which is due to that term is simply a scaling of the query point. Thus, for any iterate $x$ generated by $A$ before completing $t$ rounds of optimization, $\langle x, v_r\rangle = 0$ for all $r \geq t$ so long as the dimension is greater than the total number of queries made to $h_F$ so far plus $k + 1$; a fact which follows directly from the previous proof.

Since exactly $\lceil \frac{m}{2} \rceil$ functions are queried per round, $\sum_{i=1}^{m}\delta_{i,r} = \lfloor \frac{m}{2} \rfloor$ and thus

$$F(x) = \frac{\lambda(Q-1)}{8}\left(\langle x, v_0\rangle^2 - 2C\langle x, v_0\rangle + \zeta\langle x, v_k\rangle^2 + \sum_{r=1}^{k}\langle x, v_{r-1} - v_r\rangle^2\right) + \frac{\lambda}{2}\|x\|^2$$

where

$$Q = \frac{\lfloor \frac{m}{2} \rfloor}{m}(\frac{1}{\lambda} - 1) + 1$$

By the first order optimality conditions for $F(x)$, its optimum $x^*$ must satisfy that:

$$2\frac{Q+1}{Q-1}\langle x^*, v_0\rangle - \langle x^*, v_1\rangle = C$$

$$\langle x^*, v_{r-1}\rangle - 2\frac{Q+1}{Q-1}\langle x^*, v_r\rangle + \langle x^*, v_{r+1}\rangle = 0$$

$$\left(1 + \zeta + \frac{4}{Q-1}\right)\langle x^*, v_k\rangle - \langle x^*, v_{k-1}\rangle = 0$$

Defining $q := \frac{\sqrt{Q}-1}{\sqrt{Q}+1} < 1$ and setting $\zeta = 1 - q$, it is straightforward to confirm that

$$x^* = C\sum_{r=0}^{k}q^{r+1}v_r$$

and also that

$$F(x^*) = -\frac{\lambda C^2}{8}\left(\sqrt{Q}-1\right)^2$$

Thus, $F(0) - F(x^*) = \frac{\lambda C^2}{8}\left(\sqrt{Q}-1\right)^2$, so by choosing $C$ appropriately, we can make the initial suboptimality of our construction take any value $\epsilon_0$. Since $F$ is $\lambda$-strongly convex, for any

$x_t$, $F(x_t) - F(x^*) \geq \frac{\lambda}{2} \|x_t - x^*\|^2$. Let $x_t$ be an iterate which is generated before $t$ rounds of optimization have been completed, implying that $\langle x_t, v_r \rangle = 0$ for all $r \geq t$. So

$$\frac{F(x_t) - F(x^*)}{F(0) - F(x^*)} \geq \frac{\frac{\lambda}{2} \|x_t - x^*\|^2}{\frac{\lambda C^2}{8} \left(\sqrt{Q} - 1\right)^2}$$

$$\geq \frac{4}{C^2} \frac{C^2 \sum_{r=t}^{k} q^{2r+2}}{\left(\sqrt{Q} - 1\right)^2}$$

$$= \frac{4(q^{2t+2} - q^{2k+4})}{(1 - q^2)\left(\sqrt{Q} - 1\right)^2}$$

$$= \frac{(q^{2t} - q^{2k+2})}{\sqrt{Q}}$$

If we set $k + 1 = \left\lceil t - \frac{1}{2 \log q} \right\rceil$ then

$$\frac{F(x_t) - F(x^*)}{F(0) - F(x^*)} \geq \frac{(q^{2t} - q^{2k+2})}{\sqrt{Q}}$$

$$\geq \frac{q^{2t}}{2\sqrt{Q}}$$

$$= \frac{1}{2\sqrt{Q}} \exp\left(-2t \log \frac{1}{q}\right)$$

$$= \frac{1}{2\sqrt{Q}} \exp\left(-2t \log\left(1 + \frac{2}{\sqrt{Q} - 1}\right)\right)$$

$$\geq \frac{1}{2\sqrt{Q}} \exp\left(\frac{-4t}{\sqrt{Q} - 1}\right)$$

and when $t = \left\lfloor \frac{\sqrt{Q} - 1}{4} \log \frac{\epsilon_0}{2\sqrt{Q}\epsilon} \right\rfloor$

$$\frac{F(x_t) - F(x^*)}{F(0) - F(x^*)} \geq \frac{\epsilon}{\epsilon_0}$$

Therefore, the algorithm must complete at least $t$ rounds of queries before it can reach an $\epsilon$-suboptimal point. If $\epsilon_0 > \frac{3\epsilon}{\lambda}$ and $\lambda < \frac{1}{73}$, since each round includes at least $\frac{m}{2}$ oracle queries, this implies a lower bound of

$$\frac{m}{2} \left\lfloor \frac{\sqrt{Q} - 1}{4} \log \frac{\epsilon_0}{2\sqrt{Q}\epsilon} \right\rfloor \geq \frac{m}{40\sqrt{\lambda}} \log \frac{\epsilon_0}{2\sqrt{Q}\epsilon} \geq \frac{m}{40\sqrt{\lambda}} \log \frac{\sqrt{\lambda}\epsilon_0}{2\epsilon}$$

queries to $h_F$ in order to reach an $\epsilon$-suboptimal point. $\square$

## C   Lower bounds for randomized algorithms

To prove the deterministic lower bounds, we constructed vectors $v_r$ adversarially, orthogonalizing them to queries made by the algorithm. In the randomized setting, this is impossible, as we cannot anticipate query points. Our solution was to instead draw the important directions $v_{i,r}$ randomly in high dimensions. The intuition is that a given vector, in this case the query made by the algorithm, will have a very small inner product with a random unit vector with high probability if the dimension is large enough.

Using this fact, we construct helper functions $\psi_c$ and $\phi_c$ to replace the absolute and squared difference functions used in the deterministic lower bounds. These functions are both flat at 0 on the interval $[-c, c]$, meaning that the algorithm's query needs to have a significant inner product with $v_{i,r}$ before the oracle needs to give that vector away as a gradient or prox. We will show that each one of our constructions satisfies the following property:

**Property 1.** *For all $i$, all $t \leq k$ and $x$ such that $\forall r \geq t \,|\langle x, v_{i,r}\rangle| < \frac{c}{2}$, if $t$ is odd, then*

$$\partial f_{i,1}(x) \subseteq span\,\{x, v_{i,0}, ..., v_{i,t-1}\} \qquad and \qquad \partial f_{i,2}(x) \subseteq span\,\{x, v_{i,0}, ..., v_{i,t}\}$$

$$prox_{f_{i,1}}(x, \beta) \in span\,\{x, v_{i,0}, ..., v_{i,t-1}\} \qquad and \qquad prox_{f_{i,2}}(x, \beta) \in span\,\{x, v_{i,0}, ..., v_{i,t}\}$$

*and if $t$ is even, then*

$$\partial f_{i,1}(x) \subseteq span\,\{x, v_{i,0}, ..., v_{i,t}\} \qquad and \qquad \partial f_{i,2}(x) \subseteq span\,\{x, v_{i,0}, ..., v_{i,t-1}\}$$

$$prox_{f_{i,1}}(x, \beta) \in span\,\{x, v_{i,0}, ..., v_{i,t}\} \qquad and \qquad prox_{f_{i,2}}(x, \beta) \in span\,\{x, v_{i,0}, ..., v_{i,t-1}\}$$

In other words, when $x$ has a small inner product with $v_{i,r}$ for all $r \geq t$, then querying either $f_{i,1}$ or $f_{i,2}$ at $x$ will reveal at most $v_{i,t}$. Our bounds on the complexity of optimizing our functions are based on the principle that the algorithm can only learn one $v_{i,r}$ per query, so we need to control the probability that the hypotheses of these lemmas hold for *every* query made by the algorithm. In this section, we will bound how large the dimensionality of the problem needs to be to ensure that with high probability, only one vector is revealed to the algorithm by each oracle response.

We consider the following setup:

- For $i = 1, 2, ..., m/2$, $f_{i,1}$ and $f_{i,2}$ are pairs of component functions that satisfy Property 1 to be optimized by the randomized algorithm, we can assume that $m$ is even by letting the last component function be 0 if $m$ is odd, at the cost of a factor of $(m-1)/m$ to the complexity.

- For $i = 1, 2, ..., m/2$ and $r = 1, ..., k$, $\{v_{i,r}\}$ is a uniformly random set of **orthonormal** vectors in $\mathbb{R}^d$.

- We denote the $n^{th}$ query made by the algorithm $q^{(n)} = \left(i^{(n)}, j^{(n)}, x^{(n)}, \beta^{(n)}\right)$, which is a query to function $f_{i^{(n)}, j^{(n)}}$ at the point $x^{(n)}$ with the prox parameter $\beta^{(n)}$. We require that $\exists B$ s.t. $\left\|x^{(n)}\right\| \leq B$ for all $n$; this will be justified in the individual lower bound proofs. The $n^{th}$ query is allowed to depend on the previous $n - 1$ queries, the oracle's responses to those queries, and the randomness in the algorithm.

- For $n = 1, ..., N$, let $S_n = \text{span}\left\{x^{(t)} : t < n, \ i^{(t)} = i^{(n)}\right\}$ and let $\perp (S_n^i)$ be its orthogonal complement.

- Let $P_S v$ be the projection of the vector $v$ onto the subspace $S$, and $P_S^\perp v$ be its projection onto $\perp (S)$.

- Let $t^{(n)} = \sum_{n'=1}^{n-1} \mathbb{1}\left(i^{(n')} = i^{(n)}\right)$. The counter $t^{(n)}$ keeps track of the number of times that function $f_{i^{(n)},1}$ or $f_{i^{(n)},2}$ has been queried by the algorithm *before* the $n^{th}$ query.

- Let $U_n = \left\{v_{i^{(n)},r} : r \geq t^{(n)}\right\}$. This is the set of vectors $v_{i,r}$ which are supposed to be "unknown" to the algorithm before the $n^{th}$ query.

Ultimately, we want to prove the following statement:

$$\mathbb{P}\left(\forall n \,\forall v \in U_n \, \left|\left\langle x^{(n)}, v\right\rangle\right| < \frac{c}{2}\right) > 1 - \delta \tag{15}$$

when the dimension is adequately large. The main difficulty here is that queries made by the algorithm are allowed to depend on the oracle's responses to previous queries, which in turn depends on the vectors $v$. Therefore, there is a complicated statistical dependence between $x^{(n)}$ and $v \in U_n$ for each $n > 1$, which makes analyzing the distribution of the inner product hard. We will get around this by proving a slightly different statement, and then show that it implies (15).

Define the following "good" event:

$$G_n = \left[\forall v \in U_n \, \left|\left\langle \frac{x^{(n)}}{\|x^{(n)}\|}, P_{S_n}^\perp v\right\rangle\right| < \alpha\right]$$

where $\alpha = \frac{c}{2B(\sqrt{N}+1)}$. The following lemma shows why $G_n$ is a useful thing to look at:

**Lemma 6.** *For any $c > 0$ and $N$, $\left[\bigcap_{n=1}^N G_n\right] \implies \left[\forall n \leq N \,\forall v \in U_n \, \langle x^{(n)}, v\rangle < \frac{c}{2}\right]$.*

*Proof.* Because $\left\|x^{(n)}\right\| \le B$ and $\bigcap_{n'=1}^{n} G_{n'}$

$$\left|\left\langle x^{(n)}, v \right\rangle\right| = \left\|x^{(n)}\right\| \left|\left\langle \frac{x^{(n)}}{\|x^{(n)}\|}, P_{S_n}^\perp v \right\rangle + \left\langle \frac{x^{(n)}}{\|x^{(n)}\|}, P_{S_n} v \right\rangle\right|$$

$$\le B\left(\alpha + \sqrt{n-1}\alpha\right)$$

$$\le B\alpha + B\alpha\sqrt{N}$$

$$\le \frac{c}{2} \qquad \qquad \square$$

Now that we know that $[\forall n\ G_n]$ implies (15), we prove the following:

**Lemma 7.** *For any set of functions satisfying Property 1, any $c > 0$, any $0 < \delta < 1$, any $k$, $N$, and any dimension $d \ge \frac{32B^2 N}{c^2} \log\left(\frac{kN}{\delta}\right)$, $\mathbb{P}\left(\forall n \le N\ \forall v \in U_n\ \left|\langle x^{(n)}, v\rangle\right| < \frac{c}{2}\right) > 1 - \delta$*

*Proof.*

$$\mathbb{P}\left(\bigcap_{n=1}^{N} G_n\right) = \prod_{n=1}^{N} \mathbb{P}\left(G_n \mid G_{<n}\right)$$

where $G_{<n}$ is shorthand for $\bigcap_{n'=1}^{n-1} G_{n'}$. We lower bound each term of the product by showing that for *any* sequence of $n$ queries $q^{(1)}, ..., q^{(n)}$

$$\mathbb{P}\left(\forall v \in U_n\ \left|\left\langle \frac{x^{(n)}}{\|x^{(n)}\|}, P_{S_n}^\perp v \right\rangle\right| < \alpha \ \middle|\ G_{<n}, q^{(1)}, ..., q^{(n)}\right)$$

$$= \mathbb{P}\left(\forall v \in U_n\ \left|\left\langle \frac{P_{S_n}^\perp x^{(n)}}{\|x^{(n)}\|}, P_{S_n}^\perp v \right\rangle\right| < \alpha \ \middle|\ G_{<n}, q^{(1)}, ..., q^{(n)}\right)$$

$$> 1 - \delta'$$

Marginally, each $v$ is uniformly random on the unit sphere. Consequently, $v$ projected onto any fixed subspace is independent of the projection onto the orthogonal complement of that subspace when conditioned on the norm of the projection, so $P_{S_n}^\perp v \perp P_{S_n} v \mid \|P_{S_n} v\|$. Conditioned on $G_{<n}$, 6 and Property 1 ensures that the oracle's responses to the first $n-1$ queries were independent of $v$. So, $v$ is independent of the queries conditioned on $G_{<n}$ and $\|P_{S_t} v\|$, and those events depend only on $v$'s projection onto the subspace $S_t$. Therefore, $P_{S_n}^\perp v$ remains uniformly distributed on the sphere of radius $\sqrt{1 - \|P_{S_n} v\|^2}$ in the subspace $\perp (S_n)$. Furthermore, since the projection operator is non-expansive:

$$\mathbb{P}\left(\left|\left\langle \frac{P_{S_n}^\perp x^{(n)}}{\|x^{(n)}\|}, P_{S_n}^\perp v \right\rangle\right| \ge \alpha \ \middle|\ v \in U_n, G_{<n}, q^{(1)}, ..., q^{(n)}, \|P_{S_n} v\|\right)$$

$$< \mathbb{P}\left(\left|\left\langle \frac{P_{S_n}^\perp x^{(n)}}{\|P_{S_n}^\perp x^{(n)}\|}, \frac{P_{S_n}^\perp v}{\|P_{S_n}^\perp v\|} \right\rangle\right| \ge \alpha \ \middle|\ v \in U_n, G_{<n}, q^{(1)}, ..., q^{(n)}, \|P_{S_n} v\|\right)$$

Let $d' = \dim(\perp (S_n))$, then this is the inner product between two unit vectors, one fixed, and one uniformly random on the unit sphere in $\mathbb{R}^{d'}$. The set of vectors for which the absolute value of the inner product is greater than $\alpha$ are two "ends" of the sphere which lie above and below circles of radius $\sqrt{1 - \alpha^2}$. The total surface area of the two portions of the sphere is strictly less than the

surface area of the sphere of radius $\sqrt{1 - \alpha^2}$. Therefore,

$$\mathbb{P}\left(\left|\left\langle \frac{P^\perp_{S_n} x^{(n)}}{\|P^\perp_{S_n} x^{(n)}\|}, \frac{P^\perp_{S_n} v}{\|P^\perp_{S_n} v\|} \right\rangle\right| \geq \alpha \;\middle|\; v \in U_n, G_{<n}, q^{(1)}, ..., q^{(n)}, \|P_{S_n} v\| \right)$$

$$< \frac{\sqrt{1 - \alpha^2}^{d'-1}}{1^{d'-1}} = \left(1 - \alpha^2\right)^{\frac{d'-1}{2}}$$

and since $d' \geq d - n$ and $1 - x \leq e^{-x}$

$$\left(1 - \alpha^2\right)^{\frac{d'-1}{2}} \leq e^{-\frac{\alpha^2(d-n-1)}{2}} \leq e^{-\frac{\alpha^2(d-N-1)}{2}}$$

Using this result and a union bound over the set $U_n$ which has size at most $k$:

$$\mathbb{P}\left(\forall v \in U_n \; \left|\left\langle \frac{x^{(n)}}{\|x^{(n)}\|}, P^\perp_{S_n} v \right\rangle\right| < \alpha \;\middle|\; G_{<n}, q^{(1)}, ..., q^{(n)}, \|P_{S_n} v\| \right) > 1 - ke^{-\frac{\alpha^2(d-N-1)}{2}}$$

Since this argument applied for any $n$, any sequence of queries $q^{(1)}, ..., q^{(n)}$, and any value of $\|P_{S_n} v\|$ that is consistent with $G_{<n}$, this implies that

$$\forall n \; \mathbb{P}\left(G_n \mid G_{<n}\right) > 1 - ke^{-\frac{\alpha^2(d-N-1)}{2}}$$

$$\implies \mathbb{P}\left(\bigcap_{n=1}^N G_n\right) > \left(1 - ke^{-\frac{\alpha^2(d-N-1)}{2}}\right)^N \geq 1 - kNe^{-\frac{\alpha^2(d-N-1)}{2}}$$

So, when

$$d \geq \frac{2}{\alpha^2} \log\left(\frac{kN}{\delta}\right) + N + 1$$

then

$$\mathbb{P}\left(\bigcap_{n=1}^N G_n\right) > 1 - kNe^{-\frac{\alpha^2(d-N-1)}{2}}$$

$$\geq 1 - kNe^{-\log\frac{kN}{\delta}}$$

$$= 1 - \delta$$

Applying lemma 6 completes the proof. $\qquad\square$

Together, Lemma 7 and Property 1 allow us to ensure that any algorithm can only learn one important vector per query with high probability as long as the dimension is large enough. What is left is to show that Property 1 holds for each of our constructions and to bound the suboptimality of any iterate that has small inner product with the vectors in $U_n$.

### C.1 Non-smooth and not strongly convex components

We first consider the Lipschitz and non-strongly convex setting and prove theorem 5:

**Theorem 5.** *For any $L, B > 0$, any $0 < \epsilon < \frac{LB}{10\sqrt{m}}$, any $m \geq 2$, and any randomized algorithm $A$ with access to $h_F$, there exists a dimension $d = \mathcal{O}\left(\frac{L^4 B^6}{\epsilon^4} \log\left(\frac{LB}{\epsilon}\right)\right)$, and $m$ functions $f_i$ defined over $\mathcal{X} = \{x \in \mathbb{R}^d : \|x\| \leq B\}$, which are convex and $L$-Lipschitz continuous, such that to find a point $\hat{x}$ for which $\mathbb{E}\left[F(\hat{x}) - F(x^*)\right] < \epsilon$, $A$ must make $\Omega\big(m + \frac{\sqrt{m}LB}{\epsilon}\big)$ queries to $h_F$.*

As shown in Equations (9) and (10), we define

$$\psi_c(z) = \max\left(0, |z| - c\right)$$

and for values $b$, $c$, and $k$ to be fixed later we define $m/2$ pairs of functions, indexed by $i = 1..m/2$:

$$f_{i,1}(x) = \frac{1}{\sqrt{2}} |b - \langle x, v_{i,0} \rangle| + \frac{1}{2\sqrt{k}} \sum_{r \text{ even}}^{k} \psi_c \left( \langle x, v_{i,r-1} \rangle - \langle x, v_{i,r} \rangle \right)$$

$$f_{i,2}(x) = \frac{1}{2\sqrt{k}} \sum_{r \text{ odd}}^{k} \psi_c \left( \langle x, v_{i,r-1} \rangle - \langle x, v_{i,r} \rangle \right)$$

Assume for now that $m$ is even. If $m$ is odd, then we simply set one of the functions to 0 and the oracle complexity is reduced by a factor proportional to $\frac{m-1}{m}$.

At the end of this proof, we will show that the functions $f_{i,\cdot}$ satisfy Property 1. Since the domain, and therefore the queries made to the oracle are bounded by $B$, Property 1 and Lemma 7 ensure that when the dimension is at least $d = \frac{32B^2 N}{c^2} \log(10kN)$, for iterate $x$ generated after $N$ oracle queries, $\langle x, v_{i,r} \rangle \geq \frac{c}{2}$ for no more than $N$ vectors $v_{i,r}$ with probability $\frac{9}{10}$.

We now bound the suboptimality of $(f_{i,1} + f_{i,2})/2$ for any $x$ where $\langle x, v_{i,k} \rangle < \frac{c}{2}$.

$$\frac{1}{2}(f_{i,1}(x) + f_{i,2}(x)) = \frac{1}{2\sqrt{2}} |b - \langle x, v_{i,0} \rangle| + \frac{1}{4\sqrt{k}} \sum_{r=1}^{k} \psi_c \left( \langle x, v_{i,r-1} \rangle - \langle x, v_{i,r} \rangle \right)$$

It is straightforward to confirm that this function is minimized when $\langle x, v_{i,r} \rangle = b$ for all $r$. Since this is also true for every $i$, $F$ is minimized at $x_b = b \sum_{i=1}^{\frac{m}{2}} \sum_{r=0}^{k} v_{i,r}$. In order that $\|x_b\| = 1$ so that $x_b \in \mathcal{X}$, we set $b = \sqrt{\frac{2}{m(k+1)}}$. Thus,

$$\frac{1}{2}(f_{i,1}(x) + f_{i,2}(x)) - \frac{1}{2}(f_{i,1}(x^*) + f_{i,2}(x^*)) \geq \frac{1}{2}(f_{i,1}(x) + f_{i,2}(x)) - 0$$

$$\geq \frac{1}{2\sqrt{2}} |b - \langle x, v_{i,0} \rangle| + \frac{1}{4\sqrt{k}} \sum_{r=1}^{k} |\langle x, v_{i,r-1} \rangle - \langle x, v_{i,r} \rangle| - c$$

$$\geq -\frac{k}{4\sqrt{k}} c + \frac{1}{2\sqrt{2}} |b - \langle x, v_{i,0} \rangle| + \frac{1}{4\sqrt{k}} |\langle x, v_{i,0} \rangle - \langle x, v_{i,k} \rangle|$$

$$\geq -\frac{k}{4\sqrt{k}} c + \frac{1}{2\sqrt{2}} |b - \langle x, v_{i,0} \rangle| + \frac{1}{4\sqrt{k}} |\langle x, v_{i,0} \rangle| - \frac{1}{4\sqrt{k}} \frac{c}{2}$$

$$\geq -\frac{2k+1}{8\sqrt{k}} c + \min_{z \in \mathbb{R}} \frac{1}{2\sqrt{2}} |b - z| + \frac{1}{4\sqrt{k}} |z|$$

$$= -\frac{2k+1}{8\sqrt{k}} c + \frac{b}{4\sqrt{k}}$$

$$\geq -\frac{2k+1}{8\sqrt{k}} c + \frac{1}{4k\sqrt{m}}$$

Therefore, we set $c = \frac{\epsilon}{\sqrt{k}}$ and $k = \lfloor \frac{1}{10\epsilon\sqrt{m}} \rfloor$ so that

$$\frac{1}{2}(f_{i,1}(x) + f_{i,2}(x)) - \frac{1}{2}(f_{i,1}(x^*) + f_{i,2}(x^*)) \geq -\frac{\epsilon}{2} + \frac{5}{2}\epsilon = 2\epsilon$$

This ensures that if $\langle x, v_{i,k} \rangle < \frac{c}{2}$ for at least $m/4$ $i$'s, then $x$ cannot be $\epsilon$-suboptimal for $F$. Therefore, after $N = \frac{m(k+1)}{4}$ queries in dimension $d = \frac{32B^2 N}{c^2} \log(10kN) = \frac{32B^2}{c^2} \left( \frac{m(k+1)}{4} \right) \log \left( \frac{10mk(k+1)}{4} \right)$ then $F(x) - F(x^*) \geq \epsilon$ with probability $\frac{9}{10}$. When $\epsilon < \frac{1}{10\sqrt{m}}$, $A$ must make at least

$$\frac{m(k+1)}{4} \geq \frac{m}{4} + \frac{\sqrt{m}}{80\epsilon}$$

queries with probability $\frac{9}{10}$ Finally, we prove that Property 1 holds for our construction:

*Proof of Propetry 1 for Lipschitz, non-strongly convex construction.* First we prove the properties about the gradients:

Consider the case when $t$ is odd. From (9), it is clear that $\frac{d\psi_c}{dz}(z) = 0$ when $|z| < c$. Furthermore, for $r > t$, $|\langle x, v_{i,r-1}\rangle - \langle x, v_{i,r}\rangle| < c$. Therefore, any subgradient $g_1 \in \partial f_{i,1}(x)$ and $g_2 \in \partial f_{i,2}(x)$ can be expressed as

$$g_1 = \frac{\text{sign}(b - \langle x, v_{i,0}\rangle)}{\sqrt{2}} v_{i,0} + \frac{1}{2\sqrt{k}} \sum_{\substack{r \text{ even}}}^{t-1} \psi_c'(\langle x, v_{i,r-1}\rangle - \langle x, v_{i,r}\rangle)(v_{i,r-1} - v_{i,r})$$

$$g_2 = \frac{1}{2\sqrt{k}} \sum_{\substack{r \text{ odd}}}^{t} \psi_c'(\langle x, v_{i,r-1}\rangle - \langle x, v_{i,r}\rangle)(v_{i,r-1} - v_{i,r})$$

where $\text{sign}(0)$ can take any value in the range $[-1, 1]$ and where $\psi_c'$ is a subderivative of $\psi_c$. It is clear from these expressions that $\partial f_{i,1}(x) \subseteq \text{span}\{v_{i,0}, ..., v_{i,t-1}\}$ and $\partial f_{i,2}(x) \subseteq \text{span}\{v_{i,1}, ..., v_{i,t}\}$. The proof for the case when $t$ is even follows the same line of reasoning.

We now prove the properties about the proxs:

Since each pair of functions $f_{i,\cdot}$ operates on a separate $(k+1)$-dimensional subspace, it will be useful to decompose vectors into $x = x_i^v + x_i^\perp$ where $x_i^v = \sum_{r=0}^k \langle x, v_{i,r}\rangle v_{i,r}$ and $x_i^\perp = x - x_i^v$. First, note that

$$\text{prox}_{f_{i,1}}(x, \beta) = \arg\min_u f_{i,1}(u) + \frac{\beta}{2}\|x - u\|^2$$

$$= \arg\min_{u_i^v, u_i^\perp} f_{i,1}(u_i^v) + \frac{\beta}{2}\|x_i^v + x_i^\perp - u_i^v - u_i^\perp\|^2$$

$$= \arg\min_{u_i^v, u_i^\perp} f_{i,1}(u_i^v) + \frac{\beta}{2}\|x_i^v - u_i^v\|^2 + \|x_i^\perp - u_i^\perp\|^2$$

$$= \arg\min_{u_i^v} f_{i,1}(u_i^v) + \frac{\beta}{2}\|x_i^v - u_i^v\|^2 + \arg\min_{u_i^\perp} \frac{\beta}{2}\|x_i^\perp - u_i^\perp\|^2$$

$$= x_i^\perp + \arg\min_{u_i^v} f_{i,1}(u_i^v) + \frac{\beta}{2}\|x_i^v - u_i^v\|^2$$

$$= x_i^\perp + \text{prox}_{f_{i,1}}(x_i^v, \beta)$$

(and similarly for $f_{i,2}$). From there, the proof is similar to the proof of lemma 3. First, consider the function $f_{i,2}$ and let $t' \geq t$ be the smallest even number which is not smaller than $t$. It will be convenient to further decompose vectors into $x_i^v = x^- + x^+$ where $x^- = \sum_{r=0}^{t'-1}\langle x_i^v, v_{i,r}\rangle v_{i,r}$ and $x^+ = \sum_{r=t'}^k \langle x_i^v, v_{i,r}\rangle v_{i,r}$. So

$$f_{i,2}(x^- + x^+) = \frac{1}{2\sqrt{k}} \sum_{r \in \{1,3,...,t'-1\}} \psi_c\left(\langle x^-, v_{i,r-1}\rangle - \langle x^-, v_{i,r}\rangle\right)$$

$$+ \frac{1}{2\sqrt{k}} \sum_{r \in \{t'+1,t'+3,...k\}} \psi_c\left(\langle x^+, v_{i,r-1}\rangle - \langle x^+, v_{i,r}\rangle\right)$$

$$= f_{i,2}(x^-) + f_{i,2}(x^+)$$

Therefore,

$$\text{prox}_{f_{i,2}}(x_i^v, \beta) = \arg\min_{u^-, u^+} f_{i,2}(u^- + u^+) + \frac{\beta}{2}\|x^- + x^+ - u^- - u^+\|^2$$

$$= \arg\min_{u^-, u^+} f_{i,2}(u^- + u^+) + \frac{\beta}{2}\|x^- - u^-\|^2 + \frac{\beta}{2}\|x^+ - u^+\|^2$$

$$= \arg\min_{u^-, u^+} f_{i,2}(u^-) + f_{i,2}(u^+) + \frac{\beta}{2}\|x^- - u^-\|^2 + \frac{\beta}{2}\|x^+ - u^+\|^2$$

$$= \arg\min_{u^-} f_{i,2}(u^-) + \frac{\beta}{2}\|x^- - u^-\|^2 + \arg\min_{u^+} f_{i,2}(u^+) + \frac{\beta}{2}\|x^+ - u^+\|^2$$

Since $|\langle x^+, v_{i,r}\rangle| < \frac{c}{2}$ for all $r \geq t$, $|\langle x^+, v_{i,r-1}\rangle - \langle x^+, v_{i,r}\rangle| < c$ for $r > t$, which implies that $f_{i,2}(x^+) = 0$. Therefore, the objective of the second $\arg\min$ is non-negative and is equal to zero when $u^+ = x^+$ so

$$\text{prox}_{f_{i,2}}(x_i^v, \beta) = x^+ + \arg\min_{u^-} f_{i,2}(u^-) + \frac{\beta}{2}\left\| x^- - u^-\right\|^2$$

Therefore, when $t$ is even, $t' = t$ and $\text{prox}_{f_{i,2}}(x, \beta) \in \text{span}\{x, v_{i,0}, ..., v_{i,t-1}\}$, and when $t$ is odd, $t' = t+1$ and $\text{prox}_{f_{i,2}}(x, \beta) \in \text{span}\{x, v_{i,0}, ..., v_{i,t}\}$. A very similar line of reasoning can be used to show the statement for $f_{i,1}$. $\qquad\square$

**Remark:** As was mentioned before, Lemma 7 applies when the norm of every query point is bounded by $B$. Since all points in the domain of the optimization problem have norm bounded by $B$, this is not problematic. However, we can slightly modify our construction to make optimizing $F$ hard *even* for algorithms that are allowed to query outside of the domain.

We could redefine our functions as follows:

$$f'_{i,j}(x) = \begin{cases} f_{i,j}(x) & \|x\| \leq B \\ f_{i,j}\left(B\frac{x}{\|x\|}\right) + L\left(\|x\| - B\right) & \|x\| > B \end{cases}$$

$f'_{i,j}$ is still continuous, and $L$-Lipschitz, and it also has the property that it behaves exactly like $f_{i,j}$ on $B$-ball. However, querying the oracle of $f'_{i,j}$ outside of the $B$-ball gives no more information about the function than querying at $B\frac{x}{\|x\|}$. In fact, an algorithm that was only allowed to query within the $B$-ball would be able to simulate the oracle of $F'$. Therefore, since the algorithm that is not allowed to query at large vectors cannot optimize $F'$ quickly, and it could simulate queries with unbounded norm, it follows that querying with unbounded norm cannot improve the rate of convergence. This fact is needed in the proof of Theorem 6 below.

## C.2 Non-smooth and strongly convex components

We now prove Theorem 6 using a reduction from the Lipschitz and *non*-strongly convex setting:

**Theorem 6.** *For any $L, \lambda > 0$, any $0 < \epsilon < \frac{L^2}{200\lambda m}$, any $m \geq 2$, and any randomized algorithm $A$ with access to $h_F$, there exists a dimension $d = \mathcal{O}\left(\frac{L^4}{\lambda^3\epsilon}\log\frac{L}{\sqrt{\lambda\epsilon}}\right)$, and $m$ functions $f_i$ defined over $\mathcal{X} \subseteq \mathbb{R}^d$, which are $L$-Lipschitz continuous and $\lambda$-strongly convex, such that in order to find a point $\hat{x}$ for which $\mathbb{E}\left[F(\hat{x}) - F(x^*)\right] < \epsilon$, $A$ must make $\Omega\left(m + \frac{\sqrt{m}L}{\sqrt{\lambda\epsilon}}\right)$ queries to $h_F$.*

*Proof.* Just as in the proof of Theorem 2, we assume towards contradiction that there is an algorithm $A$ which can optimize $F$ using $o\left(m + \frac{\sqrt{m}L}{\sqrt{\lambda\epsilon}}\right)$ queries to $h_F$ in expectation. Then $A$ could be used to minimize the sum $\tilde{F}$ of $m$ functions $\tilde{f}_i$, which are convex and $L$-Lipschitz continuous over the domain $\{x : \|x\| \leq B\}$ by adding a regularizer. Let

$$F(x) = \frac{1}{m}\sum_{i=1}^{m} f_i(x) := \frac{1}{m}\sum_{i=1}^{m} \tilde{f}_i(x) + \frac{\lambda}{2}\|x\|^2$$

Note that $f_i$ is $\lambda$-strongly convex and since $\tilde{f}_i$ is $L$-Lipschitz, $f_i$ is $(L + \lambda B)$-Lipschitz continuous on the same domain. Furthermore, by setting $\lambda = \frac{\epsilon}{B^2}$,

$$\tilde{F}(x) \leq F(x) \leq \tilde{F}(x) + \frac{\epsilon}{2B^2}\|x\|^2 \leq \tilde{F}(x) + \frac{\epsilon}{2}$$

By assumption, $A$ can find an $\hat{x}$ such that $F(\hat{x}) - F(x^*) < \frac{\epsilon}{2}$ using $o\left(m + \frac{\sqrt{m}(L+\lambda B)}{\sqrt{\lambda\epsilon}}\right) = o\left(m + \frac{\sqrt{m}LB}{\epsilon}\right)$ queries to $h_F$, and

$$\frac{\epsilon}{2} > F(\hat{x}) - F(x^*) \geq \tilde{F}(\hat{x}) - \tilde{F}(\tilde{x}^*) - \frac{\epsilon}{2}$$

Thus $\hat{x}$ is $\epsilon$-suboptimal for $\tilde{F}$. However, this contradicts the conclusion of theorem 5 when $L > 0$, $\lambda > 0, 0 < \epsilon < \frac{L^2}{200\lambda m}$, and $d = \Omega\left(\frac{L^4}{\lambda^3\epsilon}\log\frac{L}{\sqrt{\lambda\epsilon}}\right)$ leads to contradiction. $\qquad\square$

## C.3 Smooth and not strongly convex components

**Theorem 7.** *For any $\gamma, B, \epsilon > 0$, any $m \geq 2$, and any randomized algorithm $A$ with access to $h_F$, there exists a sufficiently large dimension $d = \mathcal{O}\left(\frac{\gamma^2 B^6}{\epsilon^2} \log\left(\frac{\gamma B^2}{\epsilon}\right) + B^2 m \log m\right)$ and $m$ functions $f_i$ defined over $\mathcal{X} = \left\{x \in \mathbb{R}^d : \|x\| \leq B\right\}$, which are convex and $\gamma$-smooth, such that to find a point $\hat{x} \in \mathbb{R}^d$ for which $\mathbb{E}\left[F(\hat{x}) - F(x^*)\right] < \epsilon$, $A$ must make $\Omega\left(m + \sqrt{\frac{m\gamma B^2}{\epsilon}}\right)$ queries to $h_F$.*

Without loss of generality, we can assume that $\gamma = B = 1$. We will first consider the case where $\epsilon = O\left(\frac{1}{m}\right)$ and prove that $A$ must make $\Omega\left(\sqrt{\frac{m}{\epsilon}}\right)$ queries to $h_F$. Afterwards, we will show a lower bound of $\Omega(m)$ in the large-$\epsilon$ regime where that term dominates.

The function construction in this case is very similar to the non-smooth randomized construction. As in Equation (11)

$$\phi_c(z) = \begin{cases} 0 & |z| \leq c \\ 2(|z| - c)^2 & c < |z| \leq 2c \\ z^2 - 2c^2 & |z| > 2c \end{cases}$$

The key properties of this function for this proof are that it is convex, everywhere differentiable and 4-smooth, and when $|z| \leq c$, the function is constant at 0. It is also useful to note that

$$0 \leq z^2 - \phi_c(z) \leq 2c^2 \tag{16}$$

As in Equation (12), for values $a$ and $k$ to be fixed later, we define the pairs of functions for $i = 1, ..., m/2$:

$$f_{i,1}(x) = \frac{1}{16}\left(\langle x, v_{i,0}\rangle^2 - 2a\langle x, v_{i,0}\rangle + \sum_{r \in \{2,4,...\} \leq k} \phi_c\left(\langle x, v_{i,r-1}\rangle - \langle x, v_{i,r}\rangle\right)\right)$$

$$f_{i,2}(x) = \frac{1}{16}\left(\sum_{r \in \{1,3,...\} \leq k} \phi_c\left(\langle x, v_{i,r-1}\rangle - \langle x, v_{i,r}\rangle\right) + \phi_c\left(\langle x, v_{i,k}\rangle\right)\right)$$

with orthonormal vectors $v_{i,r}$ chosen randomly on the unit sphere in $\mathbb{R}^d$ as for Theorem 5.

At the end of this proof, we will show that the functions $f_{i,\cdot}$ satisfy Property 1. Since the domain, and therefore the queries made to the oracle are bounded by $B$, Property 1 and Lemma 7 ensure that when the dimension is at least $d = \frac{32B^2 N}{c^2} \log(10kN)$ then after $N$ oracle queries, $\langle x, v_{i,r}\rangle \geq \frac{c}{2}$ for no more than $N$ vectors $v_{i,r}$ with probability $\frac{9}{10}$.

Now, we will bound the suboptimality of $F_i(x) := (f_{i,1}(x) + f_{i,2}(x))/2$ at an iterate $x$ such that $|\langle x, v_{i,r}\rangle| < \frac{c}{2}$ for all $r \geq t$. From the definition of $\phi_c$:

$$F_i(x) = \frac{1}{32}\left(\langle x, v_{i,0}\rangle^2 - 2a\langle x, v_{i,0}\rangle + \sum_{r=1}^{k} \phi_c\left(\langle x, v_{i,r-1}\rangle - \langle x, v_{i,r}\rangle\right) + \phi_c\left(\langle x, v_{i,k}\rangle\right)\right)$$

$$= \frac{1}{32}\left(\langle x, v_{i,0}\rangle^2 - 2a\langle x, v_{i,0}\rangle + \sum_{r=1}^{t} \phi_c\left(\langle x, v_{i,r-1}\rangle - \langle x, v_{i,r}\rangle\right)\right)$$

$$F_i(x) \leq \frac{1}{32}\left(\langle x, v_{i,0}\rangle^2 - 2a\langle x, v_{i,0}\rangle + \sum_{r=1}^{t} \left(\langle x, v_{i,r-1}\rangle - \langle x, v_{i,r}\rangle\right)^2 + \langle x, v_{i,t}\rangle^2\right)$$

$$F_i(x) \geq \frac{1}{32}\left(\langle x, v_{i,0}\rangle^2 - 2a\langle x, v_{i,0}\rangle + \sum_{r=1}^{t} \left(\langle x, v_{i,r-1}\rangle - \langle x, v_{i,r}\rangle\right)^2 + \langle x, v_{i,t}\rangle^2\right) - \frac{t+1}{16}c^2$$

Define

$$F_i^{t+1}(x) := \frac{1}{32}\left(\langle x, v_{i,0}\rangle^2 - 2a\langle x, v_{i,0}\rangle + \sum_{r=1}^{t} \left(\langle x, v_{i,r-1}\rangle - \langle x, v_{i,r}\rangle\right)^2 + \langle x, v_{i,t}\rangle^2\right)$$

and note that in the proof of Theorem 3 we already showed that that the optimum of $F_i^t$ is achieved at

$$x_{i,t}^* = a \sum_{r=0}^{t-1} \left(1 - \frac{r+1}{t+1}\right) v_{i,r}$$

and

$$F_i^t\left(x_{i,t}^*\right) = -\frac{a^2}{32}\left(1 - \frac{1}{t+1}\right)$$

and

$$\left\|x_{i,t}^*\right\|^2 \leq \frac{a^2 t}{3}$$

Therefore, setting $a = \sqrt{\frac{6}{m(k+1)}}$ ensures that $\left\|\sum_{i=1}^{\frac{m}{2}} x_{i,k+1}^*\right\| \leq 1$. It is not necessarily true that $x^* = \sum_{i=1}^{\frac{m}{2}} x_{i,k+1}^*$, but it serves as an upper bound on the optimum.

Let $q := \lfloor \frac{k}{2} \rfloor$ and consider an iterate $x$ generated by $A$ before it makes $q-1$ queries to the functions $f_{i,1}$ and $f_{i,2}$. When $\langle x, v_{i,r} \rangle < \frac{c}{2}$ for all $r \geq q$,

$$
\begin{aligned}
F_i(x) - F_i(x^*) &\geq F_i^q(x) - \frac{qc^2}{16} - F_i(x_{i,k+1}^*) \\
&\geq F_i^q(x_{i,q}^*) - F_i^{k+1}(x_{i,k+1}^*) - \frac{qc^2}{16} \\
&= -\frac{a^2}{32}\left(1 - \frac{1}{q+1}\right) + \frac{a^2}{32}\left(1 - \frac{1}{k+2}\right) - \frac{qc^2}{16} \\
&\geq \frac{1}{32k^2 m} - \frac{kc^2}{32}
\end{aligned}
$$

where the last inequality holds as long as $k \geq 2$. When $\epsilon < \frac{1}{320m}$, setting $c = \sqrt{\frac{16\epsilon}{k}}$ and $k = \lfloor \frac{1}{\sqrt{80\epsilon m}} \rfloor \geq 2$, ensures that

$$F_i(x) - F_i(x^*) \geq \frac{5}{2}\epsilon - \frac{\epsilon}{2} = 2\epsilon$$

Therefore, if $\langle x, v_{i,r} \rangle < \frac{c}{2}$ for all $r \geq q$ is true for at least $\frac{m}{4}$ of the $i$'s, then $x$ cannot be $\epsilon$-suboptimal for $F$. So, for $N = \frac{mq}{4}$ in dimension $d = \frac{32B^2N}{c^2}\log(10kN) = \frac{32B^2}{c^2}\left(\frac{mq}{4}\right)\log(10kN)$, with probability $\frac{9}{10}$, the algorithm must make at least $\frac{mq}{4}$ queries in order to reach an $\epsilon$-suboptimal point. This gives a lower bound of

$$\frac{m}{4}q \geq \frac{\sqrt{m}}{48\sqrt{10\epsilon}}$$

which holds with probability $\frac{9}{10}$. To complete the first half of the proof, we prove that Property 1 holds for this construction:

*Proof of Property 1 for smooth and non-strongly convex construction.* First we prove the properties about gradients:

Consider the case when $t$ is odd. From equation 16, we can see that $\frac{d\phi_c}{dz}(z) = 0$ when $|z| < c$. Furthermore, for $r > t$, $|\langle x, v_{i,r-1} \rangle - \langle x, v_{i,r} \rangle| < c$. We can therefore express the gradients:

$$\nabla f_{i,1}(x) = \frac{1}{16}\left(2\langle x, v_{i,0}\rangle - 2av_{i,0} + \sum_{\substack{r=0 \\ r \text{ even}}}^{t-1} \phi_c'\left(\langle x, v_{i,r-1}\rangle - \langle x, v_{i,r}\rangle\right)\left(v_{i,r-1} - v_{i,r}\right)\right)$$

$$\nabla f_{i,2}(x) = \frac{1}{16}\left(\sum_{r \text{ odd}}^{t} \phi_c'\left(\langle x, v_{i,r-1}\rangle - \langle x, v_{i,r}\rangle\right)\left(v_{i,r-1} - v_{i,r}\right) + \phi_c'\left(\langle x, v_{i,k}\rangle\right)v_{i,k}\right)$$

It is clear from these expressions that $\nabla f_{i,1}(x) \in \text{span}\{v_{i,0}, ..., v_{i,t-1}\}$ and $\nabla f_{i,2}(x) \in \text{span}\{v_{i,0}, ..., v_{i,t}\}$. The proof for the case when $t$ is even follows the same line of reasoning.

Now, we prove the properties about proxs:

We follow the same line of reasoning as in the Lipschitz and non-strongly convex case. The only necessary addition is to show, that when $t' \geq t$ is the smallest even number which is not smaller than $t$ and $u^- = \sum_{r=0}^{t'-1} \langle u_i^v, v_{i,r} \rangle v_{i,r}$ and $u^+ = \sum_{r=t'}^{k} \langle u_i^v, v_{i,r} \rangle v_{i,r}$, then $f_{i,2}(u^- + u^+) = f_{i,2}(u^-) + f_{i,2}(u^+)$:

$$f_{i,2}(u^- + u^+) = \frac{1}{16} \Bigg( \sum_{r \in \{1,3,\ldots,t'-1\}} \phi_c \left( \langle u^-, v_{i,r-1} \rangle - \langle u^-, v_{i,r} \rangle \right)$$

$$+ \sum_{r \in \{t'+1, t'+3, \ldots\} < k} \phi_c \left( \langle u^+, v_{i,r-1} \rangle - \langle u^+, v_{i,r} \rangle \right) + \phi_c \left( \langle u^+, v_{i,k} \rangle \right) \Bigg)$$

$$= f_{i,2}(u^-) + f_{i,2}(u^+)$$

This same reasoning applies for $f_{i,1}$ or odd $t'$. $\qquad \square$

So far, we have shown a lower bound of $\Omega \left( \sqrt{\frac{m}{\epsilon}} \right)$ when $\epsilon = O \left( \frac{1}{m} \right)$. We now show a lower bound of $\Omega(m)$ for all $\epsilon > 0$, which accounts for the first term in the lower bound $\Omega \left( m + \sqrt{\frac{m}{\epsilon}} \right)$ which dominates when $\epsilon = \Omega \left( \frac{1}{m} \right)$. Consider the 0-smooth functions

$$f_i(x) = C \langle x, v_i \rangle$$

for any constant $C > 0$, and where the orthonormal vectors $v_i$ are randomly chosen as before. $F$ reaches its minimum on the unit ball at

$$\arg \min_{x : \|x\| \leq 1} F(x) = \frac{-1}{\sqrt{m}} \sum_{i=1}^{m} v_i$$

and $F(x^*) = -\frac{C}{\sqrt{m}}$. Using similar analysis as inside the proof of Lemma 7, if $d = \frac{2B^2}{\left( \frac{1}{4\sqrt{m}} \right)^2} \log 2m \leq 32 B^2 m \log 2m$ then $\mathbb{P} \left( \exists i \text{ which has not been queried s.t. } |\langle x, v_i \rangle| \geq \frac{1}{4\sqrt{m}} \right) < \frac{1}{2}$. So if fewer than $\frac{m}{2}$ functions have been queried, then with probability at least $\frac{1}{2}$:

$$F(x) - F(x^*) \geq \left( \frac{-C\sqrt{31}}{8\sqrt{m}} + \frac{-C}{8\sqrt{m}} \right) - \frac{-C}{\sqrt{m}} \geq \frac{0.16C}{\sqrt{m}}$$

so

$$\mathbb{E}\left[ F(x) - F(x^*) \right] \geq \frac{0.08C}{\sqrt{m}}$$

Therefore, by simply choosing $C = \frac{\epsilon\sqrt{m}}{0.08}$, we ensure that such a point $x$ is at least $\epsilon$-suboptimal, completing the proof for all $\epsilon > 0$.

As noted above, the queries made to the oracle must be bounded for Lemma 7. Since the domain of $F$ is the $B$-ball, this is easy to satisfy. If we want to ensure that our construction is still hard to optimize, *even* if the algorithm is allowed to query arbitrarily large vectors, then we can modify our construction in the following way;

$$f'_{i,j}(x) = \begin{cases} f_{i,j}(x) & \|x\| \leq B \\ f_{i,j}\left( B \frac{x}{\|x\|} \right) + \left\langle \nabla f_{i,j}\left( B \frac{x}{\|x\|} \right), x - B \frac{x}{\|x\|} \right\rangle & \|x\| > B \end{cases}$$

This function is continuous and smooth, and also has the property that querying the oracle at a point $x$ outside of the $B$-ball is cannot be more informative than querying at $B \frac{x}{\|x\|}$. That is, an algorithm that is not allowed to query outside the $B$-ball can simulate such queries using its restricted oracle. Since this restricted algorithm cannot optimize quickly, but can still calculate the oracle outputs that it would have recieved by querying large vectors, it follows that an unrestricted algorithm could not optimize this function quickly either.

**Remark:** Another variant of (1) that one might consider is an unconstrained optimization problem, where we assume that the minimizer of $F$ lies on the interior of that ball. In other words, we could consider a version of (1) where the gradient of $F$ must vanish on the interior of $\mathcal{X}$.

In this case, there is little reason to consider any $\epsilon$ larger than $\frac{\gamma B^2}{2}$, since $F(0) - F(x^*) \leq \frac{\gamma B^2}{2}$ always (by smoothness $F(0) - F(x^*) \leq \langle \nabla F(x^*), x_0 - x^* \rangle + \frac{\gamma}{2} \|x^*\|^2 \leq \frac{\gamma B^2}{2}$). Consequently, when $\epsilon \geq \frac{\gamma B^2}{2}$ there is a trivial upper bound of *zero* oracle queries, as just returning the zero vector guarantees $\epsilon$-suboptimality. We can construct functions so that Theorem 7 still applies for $0 < \epsilon < \frac{9\gamma B^2}{128}$. In the previous proof, the first construction is still valid in the unconstrained case since the minimizer lies within the unit ball. For the $\Omega(m)$ term, consider the 1-smooth functions (assume w.l.o.g. that $\gamma = B = 1$)

$$f_i(x) = \sqrt{m} \, \langle x, \, v_i \rangle + \frac{\|x\|^2}{2}$$

where the $m$ orthonormal vectors $v_i$ are drawn randomly from the unit sphere in $\mathbb{R}^d$ as in the previous construction. The gradient of $F$ vanishes at $x^* = -\frac{1}{\sqrt{m}} \sum_{i=1}^m v_i$, (note $\|x^*\| = 1$) and $F(x^*) = -\frac{1}{2}$. Using similar techniques as inside the proof of Lemma 7 if $d = \frac{2B^2}{\left(\frac{1}{4\sqrt{m}}\right)^2} \log 10m \leq 32B^2 m \log 10m$, then for any iterate $x$ generated by $A$ before $f_i$ has been queried, $\mathbb{P}\left(\exists i \text{ which has not been queried s.t. } |\langle x, \, v_i \rangle| \geq \frac{1}{4\sqrt{m}}\right) < \frac{9}{10}$. Furthermore, if $|\langle x, \, v_i \rangle| < \frac{1}{4\sqrt{m}}$ for more than $\frac{m}{2}$ of the functions, then

$$
\begin{aligned}
F(x) &= \frac{1}{m} \sum_{i=1}^m \sqrt{m} \, \langle x, \, v_i \rangle + \frac{\|x\|^2}{2} \\
&\geq \frac{1}{m} \left( \frac{m}{2} \cdot \sqrt{m} \frac{-1}{\sqrt{m}} + \frac{m}{2} \cdot \sqrt{m} \frac{-1}{4\sqrt{m}} \right) + \frac{\frac{m}{2} \cdot \frac{1}{m} + \frac{m}{2} \cdot \frac{1}{16m}}{2} \\
&= \frac{-23}{64}
\end{aligned}
$$

Therefore, if fewer than $\frac{m}{2}$ functions have been queried, then with probability at least $\frac{9}{10}$:

$$F(x) - F(x^*) \geq \frac{-23}{64} - \frac{-1}{2} = \frac{9}{64}$$

so

$$\mathbb{E}\left[F(x) - F(x^*)\right] \geq \frac{9}{128}$$

This proves a lower bound of $\Omega(m)$ for $0 < \epsilon < \frac{9\gamma B^2}{128}$.

### C.4  Smooth and strongly convex components

In the smooth and strongly convex case, we cannot use the same simple reduction that was used to prove Theorem 6. Using that construction, we would be able to show a lower bound of $m$, but would not be able to show any dependence on $\epsilon$, so the lower bound would be loose. Instead, we will use an explicit construction similar to the one used in Theorem 7.

**Theorem 8.** *For any $m \geq 2$, any $\gamma, \lambda > 0$ such that $\frac{\gamma}{\lambda} > 161m$, any $\epsilon > 0$, any $\epsilon_0 > 60\epsilon\sqrt{\frac{\gamma}{\lambda m}}$, and any randomized algorithm $A$, there exists a dimension $d = \mathcal{O}\left(\frac{\gamma^{2.5}\epsilon_0}{\lambda^{2.5}\epsilon} \log^3\left(\frac{\lambda\epsilon_0}{\gamma\epsilon}\right) + \frac{m\gamma\epsilon_0}{\lambda\epsilon} \log m\right)$, domain $\mathcal{X} \subseteq \mathbb{R}^d$, $x_0 \in \mathcal{X}$, and $m$ functions $f_i$ defined on $\mathcal{X}$ which are $\gamma$-smooth and $\lambda$-strongly convex, and such that $F(x_0) - F(x^*) = \epsilon_0$ and such that in order to find a point $\hat{x} \in \mathcal{X}$ such that $\mathbb{E}[F(\hat{x}) - F(x^*)] < \epsilon$, $A$ must make $\Omega\left(m + \sqrt{\frac{m\gamma}{\lambda}} \log\left(\frac{\epsilon_0}{\epsilon}\sqrt{\frac{m\lambda}{\gamma}}\right)\right)$ queries to $h_F$.*

*Proof.* We will prove the theorem for a 1-smooth, $\lambda$-strongly convex problem, for $\lambda < \frac{1}{73m}$, which can be generalized by scaling.

As in the proof for the non-strongly convex case, we introduce the 4-smooth helper function

$$\phi_c(z) = \begin{cases} 0 & |z| \le c \\ 2(|z| - c)^2 & c < |z| \le 2c \\ z^2 - 2c^2 & |z| > 2c \end{cases}$$

using which we will construct $m/2$ pairs of functions, which will each be based on the following. As in previous proofs, we randomly select orthonormal vectors $v_{i,r}$ from $\mathbb{R}^d$. Then, for constants $k$, $C$, and $\zeta$ to be decided upon later; with $\tilde{\lambda} := m \cdot \lambda$; and for $i = 1, ..., \lfloor m/2 \rfloor$ define the following pairs of functions (if $m$ is odd, let $f_m(x) = \frac{\tilde{\lambda}}{2m} \|x\|^2$):

$$f_{i,1}(x) = \frac{1 - \tilde{\lambda}}{16} \left( \langle x, v_{i,0} \rangle^2 - 2C \langle x, v_{i,0} \rangle \sum_{r \text{ even}}^{k} \phi_c \left( \langle x, v_{i,r-1} \rangle - \langle x, v_{i,r} \rangle \right) \right) + \frac{\tilde{\lambda}}{2m} \|x\|^2$$

$$f_{i,2}(x) = \frac{1 - \tilde{\lambda}}{16} \left( \zeta\phi_c(\langle x, v_{i,k} \rangle) + \sum_{r \text{ odd}}^{k} \phi_c \left( \langle x, v_{i,r-1} \rangle - \langle x, v_{i,r} \rangle \right) \right) + \frac{\tilde{\lambda}}{2m} \|x\|^2$$

When $\tilde{\lambda} \in [0, 1]$ these function are 1-smooth and $\lambda$-strongly convex.

These functions also have Property 1, but we will omit the proof, as it follows directly from the proof in Appendix C.3. Intuitively, the squared norm reveals no new information about the vectors $v_{i,r}$ besides what is already included in the query point $x$.

When all of the queries are bounded by $B$, Property 1 along with Lemma 7 ensures that when $d = \frac{32B^2N}{c^2} \log(10kN)$, after the algorithm make $N$ queries $\langle x, v_{i,r} \rangle \ge \frac{c}{2}$ for at most $N$ of the vectors $v_{i,r}$ with probability $\frac{9}{10}$. For this probability bound to apply, we need that all of the queries made by the algorithm are within a $B$-ball around the origin. We know that $F(0) - F(x^*) = \epsilon_0$, and by strong-convexity $F(0) \ge F(x^*) + \frac{\lambda}{2} \|x^*\|^2$, therefore, $\|x^*\| \le \sqrt{\frac{2\epsilon_0}{\lambda}} =: B$. Since the optimum point must lie in the $B$-ball around the origin, we will restrict the algorithm to query only at points within the $B$-ball. At the end of the proof, we will show that with a small modification to the functions *outside* of the $B$-ball, querying at vectors of large norm cannot help the algorithm.

Now it remains to lower bound the suboptimality of the pair $f_{i,1}$ and $f_{i,2}$ at an iterate which is orthogonal to all vectors $v_{i,r}$ for $r > t$:

In order to bound the suboptimality of a pair of functions $i$, it will be convenient to bundle up all of the terms which affect the value of $\langle x^*, v_{i,r} \rangle$ from all $m$ of the component functions. Most of those terms are contained in $f_{i,1}$ and $f_{i,2}$, however, $\|x\|^2$ terms in each of the other components also affect the value of $\langle x^*, v_{i,r} \rangle$. For each $i$, consider the projection operator $P_i$ which projects a vector $x$ onto the subspace spanned by $\{v_{i,r}\}_{r=0}^{k}$, and $P_\perp$ projecting onto the space orthogonal to $v_{i,r}$ for all $i, r$. Now decompose

$$\frac{\tilde{\lambda}}{2m} \|x\|^2 = \frac{\tilde{\lambda}}{2m} \left( \sum_{i=1}^{\lfloor \frac{m}{2} \rfloor} \|P_i x\|^2 + \|P_\perp x\|^2 \right)$$

Gather all $m$ of the $\frac{\tilde{\lambda}}{2m} \|P_i x\|^2$ terms and split them amongst $f_{i,1}$ and $f_{i,2}$ to make the following modified functions:

$$\tilde{f}_{i,1}(x) = f_{i,1} - \frac{\tilde{\lambda}}{2m} \|x\|^2 + \frac{\tilde{\lambda}}{4} \|P_i x\|^2$$

$$\tilde{f}_{i,2}(x) = f_{i,2} - \frac{\tilde{\lambda}}{2m} \|x\|^2 + \frac{\tilde{\lambda}}{4} \|P_i x\|^2$$

After this shuffle, all of the terms affecting $\langle x^*, v_{i,r} \rangle$ are contained in these two functions which will help the analysis. Note that there is also a remaining $\frac{\tilde{\lambda}}{2} \|P_\perp x\|^2$ term, however, this term is not very important to track since we are bounding the suboptimality of $F$, which can only increase by

considering that non-negative term and $P_\perp x^* = \vec{0}$. Now, consider

$$\frac{1}{2}\left(\tilde{f}_{i,1}(x) + \tilde{f}_{i,2}(x)\right) = \frac{1-\tilde{\lambda}}{32}\Bigg(\langle x,\, v_{i,0}\rangle^2 - 2C\langle x,\, v_{i,0}\rangle + \zeta\phi_c(\langle x,\, v_{i,k}\rangle)$$

$$+ \sum_{r=1}^{k}\phi_c\left(\langle x,\, v_{i,r-1}\rangle - \langle x,\, v_{i,r}\rangle\right)\Bigg) + \frac{\tilde{\lambda}}{4}\|P_i x\|^2$$

If we define

$$F_i^t(x) := \frac{1-\tilde{\lambda}}{32}\Bigg(\langle x,\, v_{i,0}\rangle^2 - 2C\langle x,\, v_{i,0}\rangle + \langle x,\, v_{i,t}\rangle^2$$

$$+ \sum_{r=1}^{t}\left(\langle x,\, v_{i,r-1}\rangle - \langle x,\, v_{i,r}\rangle\right)^2\Bigg) + \frac{\tilde{\lambda}}{4}\|P_i x\|^2$$

and

$$F_i(x) := \frac{1-\tilde{\lambda}}{32}\Bigg(\langle x,\, v_{i,0}\rangle^2 - 2C\langle x,\, v_{i,0}\rangle + \zeta\langle x,\, v_{i,k}\rangle^2$$

$$+ \sum_{r=1}^{k}\left(\langle x,\, v_{i,r-1}\rangle - \langle x,\, v_{i,r}\rangle\right)^2\Bigg) + \frac{\tilde{\lambda}}{4}\|P_i x\|^2$$

then when $|\langle x,\, v_{i,r}\rangle| < \frac{c}{2}$

$$F_i^t(x) \le \frac{1}{2}\left(\tilde{f}_{i,1}(x) + \tilde{f}_{i,2}(x)\right) + \frac{(1-\tilde{\lambda})(t+1)}{16}c^2$$

and for any $y$

$$\frac{1}{2}\left(\tilde{f}_{i,1}(y) + \tilde{f}_{i,2}(y)\right) \le F_i(y)$$

and, conveniently, $F_i$ is very similar to the construction from Appendix B.4. In particular, let $\tilde{Q} := \frac{1}{2}(\frac{1}{\tilde{\lambda}} - 1) + 1$, then

$$F_i^t(x) = \frac{1}{2}\left(\frac{\tilde{\lambda}(\tilde{Q}-1)}{8}\Bigg(\langle x,\, v_{i,0}\rangle^2 - 2C\langle x,\, v_{i,0}\rangle + \langle x,\, v_{i,t}\rangle^2\right.$$

$$\left. + \sum_{r=1}^{t}\left(\langle x,\, v_{i,r-1}\rangle - \langle x,\, v_{i,r}\rangle\right)^2\Bigg) + \frac{\tilde{\lambda}}{2}\|P_i x\|^2\right)$$

$$F_i(x) = \frac{1}{2}\left(\frac{\tilde{\lambda}(\tilde{Q}-1)}{8}\Bigg(\langle x,\, v_{i,0}\rangle^2 - 2C\langle x,\, v_{i,0}\rangle + \zeta\langle x,\, v_{i,k}\rangle^2\right.$$

$$\left. + \sum_{r=1}^{k}\left(\langle x,\, v_{i,r-1}\rangle - \langle x,\, v_{i,r}\rangle\right)^2\Bigg) + \frac{\tilde{\lambda}}{2}\|P_i x\|^2\right)$$

We have already showed in Appendix B.4 that if $\hat{x} := \arg\min_x F_i(x)$, and if

$$C > \frac{12\sqrt{\epsilon}}{\tilde{\lambda}(\sqrt{\tilde{Q}}-1)} \implies 2\epsilon_0^i := 2\left(F_i(0) - F_i(\hat{x})\right) > \frac{30\epsilon}{\tilde{\lambda}}$$

$$\zeta = \frac{2}{\sqrt{\tilde{Q}}+1}$$

$$\tilde{\lambda} < \frac{1}{73}$$

$$t = \left\lfloor \frac{\sqrt{\tilde{Q}}-1}{4}\log\frac{\epsilon_0^i}{20\sqrt{\tilde{Q}\epsilon}}\right\rfloor$$

$$|\langle x,\, v_{i,r}\rangle| \le \frac{c}{2}\quad \forall r > t$$

then,

$$2\left(F_i^t(x) - F_i(\hat{x})\right) \geq 10\epsilon$$

Therefore,

$$
\begin{aligned}
10\epsilon &\leq 2\left(F_i^t(x) - F_i(\hat{x})\right) \\
&\leq 2\left(\frac{1}{2}\left(\tilde{f}_{i,1}(x) + \tilde{f}_{i,2}(x)\right) + \frac{(1-\tilde{\lambda})(k+\zeta)}{16}c^2 - \frac{1}{2}\left(\tilde{f}_{i,1}(\hat{x}) + \tilde{f}_{i,2}(\hat{x})\right)\right) \\
&\leq \left(\tilde{f}_{i,1}(x) + \tilde{f}_{i,2}(x)\right) + \frac{(1-\tilde{\lambda})(k+\zeta)}{8}c^2 - \left(\tilde{f}_{i,1}(x^*) + \tilde{f}_{i,2}(x^*)\right)
\end{aligned}
$$

So

$$\left(\tilde{f}_{i,1}(x) + \tilde{f}_{i,2}(x)\right) - \left(\tilde{f}_{i,1}(x^*) - \tilde{f}_{i,2}(x^*)\right) \geq 10\epsilon - \frac{(1-\tilde{\lambda})(k+\zeta)}{8}c^2$$

Setting

$$c = \sqrt{\frac{16\epsilon}{(1-\tilde{\lambda})(k+\zeta)}}$$

then

$$\left(\tilde{f}_{i,1}(x) + \tilde{f}_{i,2}(x)\right) - \left(\tilde{f}_{i,1}(x^*) - \tilde{f}_{i,2}(x^*)\right) \geq 10\epsilon - \frac{(1-\tilde{\lambda})(k+\zeta)}{8}c^2 = 8\epsilon$$

Therefore, if at least $m/4$ of the pairs $i$ it holds that $|\langle x, v_{i,r}\rangle| < \frac{c}{2}$ for $r > t$, then

$$
\begin{aligned}
F(x) - F(x^*) &\geq \frac{1}{m}\sum_{i=1}^{\lfloor\frac{m}{2}\rfloor}\left(\tilde{f}_{i,1}(x) + \tilde{f}_{i,2}(x)\right) - \left(\tilde{f}_{i,1}(x^*) - \tilde{f}_{i,2}(x^*)\right) \\
&\geq \frac{m}{4} \cdot \frac{1}{m} \cdot 8\epsilon \\
&= 2\epsilon
\end{aligned}
$$

As a consequence of this, when the dimension is $d = \frac{32}{c^2}\left(\frac{2\epsilon_0}{\lambda}\right)\left(\frac{mt}{4}\right)\log\left(\frac{5kmt}{2}\right)$ then with probability $\frac{9}{10}$ the optimization algorithm must make at least $t$ queries to each of at least $\frac{m}{4}$ pairs of functions in order to reach an $\epsilon$-suboptimal solution in expectation. So, when

$$
\begin{aligned}
\lambda &\leq \frac{1}{161m} \\
\frac{\epsilon_0}{\epsilon} &\geq \frac{60}{\sqrt{m\lambda}}
\end{aligned}
$$

this gives a lower bound of

$$\left\lceil \frac{m}{4} \right\rceil \cdot t \geq \frac{m}{4} \left\lfloor \frac{\sqrt{\tilde{Q}}-1}{4} \log \frac{\epsilon_0^i}{20\sqrt{\tilde{Q}}\epsilon} \right\rfloor$$

$$\geq \frac{m}{4} \left\lfloor \frac{\sqrt{\tilde{Q}}-1}{4} \log \frac{\epsilon_0}{40\sqrt{\tilde{Q}}\epsilon} \right\rfloor$$

$$\geq \frac{m}{4} \frac{\sqrt{\tilde{Q}}-1}{8} \log \frac{\epsilon_0}{40\sqrt{\tilde{Q}}\epsilon}$$

$$\geq \frac{m}{4} \frac{3}{40\sqrt{m\lambda}} \log \frac{\epsilon_0\sqrt{m\lambda}}{30\epsilon}$$

$$= \frac{3}{160} \sqrt{\frac{m}{\lambda}} \log \frac{\epsilon_0\sqrt{m\lambda}}{30\epsilon}$$

$$= \Omega\left(\sqrt{\frac{m}{\lambda}} \log \frac{\epsilon_0\sqrt{m\lambda}}{\epsilon}\right)$$

The same argument as was used in the discussion after theorem 7 to show the $\Omega(m)$ term of the lower bound can be used here, as the function in that construction was both smooth and strongly convex.

As mentioned above, Lemma 7 requires that the norm of all query points be bounded by $B$. We argued above that the optimum of $F$ must lie within the $B$-ball around the origin. Even so, we can slightly modify our construction to show that even if the algorithm were allowed to query arbitrarily large points, it still would not be able to optimize $F$ quickly. Define:

$$f'_{i,j}(x) = \begin{cases} f_{i,j}(x) & \|x\| \leq B \\ f_{i,j}\left(B\frac{x}{\|x\|}\right) + \left\langle \nabla f_{i,j}\left(B\frac{x}{\|x\|}\right), x - B\frac{x}{\|x\|} \right\rangle + \frac{\lambda}{2} \left\|x - B\frac{x}{\|x\|}\right\|^2 & \|x\| > B \end{cases}$$

This new function is continuous, $\gamma$-smooth, and $\lambda$-strongly convex, and it also has the property that querying the function at a point $x$ outside the $B$-ball, it is no more informative than querying at $B\frac{x}{\|x\|}$. That is, an algorithm that was not allowed to query outside the $B$-ball could simulate the result of such queries. Since that restricted algorithm can't optimize $F'$ well, as proven above, another algorithm which could query at arbitrary points, therefore could not either. □

### C.5  Non-smooth components when $\epsilon$ is large

**Theorem 11.** *For any $L, B > 0$, any $\frac{10}{\sqrt{m}} < \epsilon < \frac{1}{4}$, and any $m \geq 161$, there exists $m$ functions $f_i$ which are convex and $L$-Lipschitz continuous defined on $\mathcal{X} = \{x \in \mathbb{R} : |x| \leq B\}$ such that for any randomized algorithm $A$ for solving problem (1) using access to $h_F$, $A$ must make at least $\Omega\left(\frac{L^2B^2}{\epsilon^2}\right)$ queries to $h_F$ in order to find a point $\hat{x}$ such that $\mathbb{E}[F(\hat{x}) - F(x^*)] < \epsilon$.*

*Proof.* Without loss of generality, we can assume that $L = B = 1$. We construct $m$ functions $f_i$ on $\mathbb{R}^1$ in the following manner: first, sample $p$ from the following distribution

$$p = \begin{cases} \frac{1}{2} - 2\epsilon & \text{w.p. } \frac{1}{2} \\ \frac{1}{2} + 2\epsilon & \text{w.p. } \frac{1}{2} \end{cases}$$

Then for $i = 1, ..., m$, we define

$$f_i(x) = \begin{cases} x & \text{w.p. } p \\ -x & \text{w.p. } 1-p \end{cases}$$

Consider now the task of optimizing $F(x) = \frac{1}{m}\sum_{i=1}^m f_i(x) = \frac{Yx}{m}$. Clearly, $F$ is optimized at $-\text{sign}(Y)$, and as long as $|Y| > 2m\epsilon$, then any $x$ which is $\epsilon$-suboptimal given $\text{sign}(Y) = +1$

must be at least $3\epsilon$-suboptimal given $\text{sign}(Y) = -1$. Using Chernoff bounds, $\mathbb{P}(|Y| \leq 2m\epsilon) \leq \exp(-\frac{\epsilon^2 m}{2}) < \exp(-5)$. Therefore, since the expected suboptimality of an iterate $x$ is at least

$$
\begin{aligned}
\mathbb{E}\left[F(x) - F(x^*)\right] &\geq 3\mathbb{P}\left(\text{sign}(x) \neq -\text{sign}(Y)\||Y| \geq 2m\epsilon\right)\mathbb{P}\left(|Y| \geq 2m\epsilon\right)\cdot\epsilon \\
&> 3(1 - \exp(-5))\mathbb{P}\left(\text{sign}(x) \neq -\text{sign}(Y)\||Y| \geq 2m\epsilon\right)\cdot\epsilon
\end{aligned}
$$

Therefore, until the algorithm has made enough queries so that

$$
\mathbb{P}\left(\text{sign}(x) \neq -\text{sign}(Y)\||Y| \geq 2m\epsilon\right) < \frac{1}{3 - 3\exp(-5)}
$$

the expected suboptimality is greater than $\epsilon$. By a standard information theoretic result [2, 19], achieving that probability of success at predicting the sign of $Y$ implies a comparable level of accuracy at distinguishing between $p = 0.5 + 2\epsilon$ and $p = 0.5 - 2\epsilon$, and that requires at least $\frac{1}{128\epsilon^2}$ queries to $h_F$. $\qquad\square$

It is straightforward to show a lower bound of $\Omega\left(\frac{L^2}{\lambda\epsilon}\right)$ for strongly convex functions using the same reduction by regularization as in the proofs of theorems 2 and 6. We also note that this lower bound implies a lower bound of $\Omega(m)$ for smooth functions, whether strongly convex or not. Each function $f_i$ in this construction is linear, and therefore is trivially 0-smooth. We make the gradient of each function arbitrarily large by multiplying each $f_i$ by a large number. As the multiplier grows, the algorithm need be more and more certain of the sign of $Y$ in order to achieve a expected suboptimality of less than $\epsilon$. Thus for a sufficiently large multiplier, the algorithm must query $\Omega(m)$ functions. We cannot force it to query more than that, of course, since it only needs to query $m$ functions to know the sign of $Y$ with probability 1.