[Reviews · NeurIPS 2016]

Reviewer 1

Summary

The authors establish deterministic and stochastic minimax bounds for minimizing sums of functions in the case where (sub)gradient of prox-oracle is available for each component. The paper is well written and the results seem highly non-trivial. As a results, I would recommend its acceptance.

Qualitative Assessment

The results are very sound but will be of interest for a limited group of expert in composite optimization algorithms. Note that the proposed approach to bound construction has several drawbacks and leave space for improvement. For instance, the construction of stochastic bound assumes that the problem dimension is much higher than the total number of iterations and the number of terms in the sum. In other words, there is an important part of the range of problem dimensions d for which minimax rates remain not understood. On the other hand, construction of the deterministic lower bound assumes specific querying policy. Another limitation is that the prox computation in the case of the functional family used in the construction of the lower bound is NOT easy (yet, the results of the paper do not say much about complexity of the corresponding prox computation). I also have few minor comments concerning the presentation. 1) In the introduction some confusion stems from the repeated use of term “gradient” in the non-smooth context (e.g. l. 33-41, etc). This should be fixed. 2) L. 66: “…algorithms: The…” 3) L. 159: v_r is orthogonal to what? What is the exact sense of “query” (seems that v_r should be orthogonal to all previous search points and v_k, k=0,…,r-1) 4) L. 314: “… resolving an question…” 5) The word “othonormal” appears several times (l. 158, 212)

Confidence in this Review

2-Confident (read it all; understood it all reasonably well)


Reviewer 2

Summary

The authors presented new lower complexity bound that can cover more algorithms and/or problems. The paper is well-written and interesting.

Qualitative Assessment

If I do not miss anything, there was a small mistake in the authors' statement on the lower bound in [1]. The bound in [1] was \sqrt{m \gamma/\lambda} \log (1/\epsilon) +m, while the lower bound obtained in [8] was slightly stronger, i.e., (\sqrt{m \gamma/\lambda} +m) \log (1/\epsilon). An upper complexity bound, which matches the latter lower complexity bound was also presented in [8].

Confidence in this Review

2-Confident (read it all; understood it all reasonably well)


Reviewer 3

Summary

The paper provides tight upper and lower bounds on the complexity of minimizing the average of m convex functions using gradient and prox oracles of the component functions.

Qualitative Assessment

It is a strong, very well written paper, providing in-depth analysis of the complexity bounds. The authors have shown that randomization can reduce complexity, which is a neat result.

Confidence in this Review

1-Less confident (might not have understood significant parts)


Reviewer 4

Summary

The authors provide some complexity bounds on problems that are interested in optimizing composite functions of the form F = \frac{1}{m} \sum_{i=1}^{n} f_i when one knows 0th and 1st order terms: f_i(x),\nabla f_i(x) and \Prox_{f_i}(x,.). Then, the paper produces a very long tiresome results on what can be said on the complexity of optimization procedures in several different frameworks (smooth, non smooth, deterministic, stochastic, upper bounds, lower bounds). The paper then proposes a very (very) long supplementary material that gathers bibliographic comments and proofs of the proposed results.

Qualitative Assessment

I am not able, from a theoretical point of view, to produce a reliable judgement of the paper. However, I can comment several drawbacks: 1) If I understand the global motivation of saying something in each situation for a journal paper, for a brief contribution in a conference proceedings, such a work is very painful for the referee. If you imagine I have to read 7 papers like this one in one month, you can easily conclude that it is not possible for me to produce a constructive reading of this work: 28 pages of a very very NOT self-contained paper, with an excessive abuse of acronysms and without any motivation in the beginning of the introduction. It results in a very boring enumeration of a potpourri of results. 2) The paper is not motivated at any moment. It lacks of real focus on a particular motivated example. Please, reduce the size of the supplementary material and just prove one or two result with a clear example, a clear simulation, a clear state of the art and a clear discussion. The paper does not contain at least one of this point above. 3) The way the paper is written is far from being standard for a mathematician: for example, the lines from 101 to 106 are not written rigorously (even though I believe the results being true). Minor comments: Line 115: strange sentence. Line 123: "Using the Lemma allows us to" -> change the sentence Line 123-135: I am not fan of this paragraph, with an enumeration of non horizon free methods: most of penalty terms involve highly dependent terms with respect to generally unkwnown parameters. Line 158: othonormal Line 161: the following truncations of EQUATION (6) Supp material: reduce the 20 pages to 3/4 pages!!!

Confidence in this Review

1-Less confident (might not have understood significant parts)


Reviewer 5

Summary

This paper proved several lower bounds for iterations on both smooth and non-smooth convex optimization problems with finite sum. Specifically, the proofs cover the class of proximal stochastic algorithms. Further, a comprehensive survey on current upper and lower bounds is presented. The authors actually found a systematic method to construct the worst case scenarios for smooth/non-smooth with bounded/strongly convex objective functions.

Qualitative Assessment

The authors present a survey of upper and lower bounds for finite-sum convex optimization problems and introduce a systematic methods to construct the worst case scenarios. However, these bounds only hold when the variable dimension d >> number of functions m (e.g., Thm 2), so that the # of iterations k <= d (e.g., Thm 2). For randomized algorithm, the d/k may even be as loose as O(L^2/eps^2) (e.g., Thm .5). Thus, the lower bound is more about the convergence of the first few iterations. Minor comments: 1. (Appendix line 479) Instead of "all subgradient has the form", I think it is "all subgradient contains the form". 2. (Appendix line 507) \tilde{x}^* is not defined, and I don't understand the second inequality. 3. (Appendix line 529) F^t(x) = ... = \floor{m/2} 4. (Appendix line 536) a^2 instead of a 5. (Appendix line 605) It should be d\geq 2B^2/c^2\log(1/\delta), instead of B^2/2c^2\log(...). Because (1-c^2/B^2)^{d/2} <= e^{-c^2/B^2 * d/2} <= e^{-c^2/B^2 *1/2* B^2/2c^2 \log(1/\delta)} = e^{-1/4 *\log(1/\delta)} \neq \delta

Confidence in this Review

1-Less confident (might not have understood significant parts)